# Deep neural networks enable quantitative movement analysis using single-camera videos

Łukasz Kidziński [1,4]✉, Bryan Yang[1,4], Jennifer L. Hicks[1], Apoorva Rajagopal[1], Scott L. Delp[1] & Michael H. Schwartz[2,3]✉

Many neurological and musculoskeletal diseases impair movement, which limits people's function and social participation. Quantitative assessment of motion is critical to medical decision-making but is currently possible only with expensive motion capture systems and highly trained personnel. Here, we present a method for predicting clinically relevant motion parameters from an ordinary video of a patient. Our machine learning models predict parameters include walking speed ($r = 0.73$), cadence ($r = 0.79$), knee flexion angle at maximum extension ($r = 0.83$), and Gait Deviation Index (GDI), a comprehensive metric of gait impairment ($r = 0.75$). These correlation values approach the theoretical limits for accuracy imposed by natural variability in these metrics within our patient population. Our methods for quantifying gait pathology with commodity cameras increase access to quantitative motion analysis in clinics and at home and enable researchers to conduct large-scale studies of neurological and musculoskeletal disorders.

[1] Department of Bioengineering, Stanford University, Stanford, CA 94305, USA. [2] Center for Gait and Motion Analysis, Gillette Children's Specialty Healthcare, St. Paul, MN 55101, USA. [3] Department of Orthopedic Surgery, University of Minnesota, Minneapolis, MN 55454, USA. [4] These authors contributed equally: Łukasz Kidziński, Bryan Yang. ✉email: lukasz.kidzinski@stanford.edu; schwa021@umn.edu

Gait metrics, such as walking speed, cadence, symmetry, and gait variability are valuable clinical measurements in conditions such as Parkinson's disease[1], osteoarthritis[2], stroke[3], cerebral palsy[4], multiple sclerosis[5], and muscular dystrophy[6]. Laboratory-based optical motion capture is the current gold standard for clinical motion analysis (Fig. 1a); it is used to diagnose pathological motion, plan treatment, and monitor outcomes. Unfortunately, economic and time constraints inhibit the routine collection of this valuable, high-quality data. Further, motion data collected in a laboratory may fail to capture how individuals move in natural settings. Recent advances in machine learning, along with the ubiquity and low cost of wearable sensors and smartphones, have positioned us to overcome the limitations of laboratory-based motion analysis. Researchers have trained machine learning models to estimate gait parameters[7,8] or detect the presence of disease[9], but current models often rely on data generated by specialized hardware such as optical motion capture equipment, inertial measurement units, or depth cameras[10,11].

Standard video has the potential to be a low-cost, easy-to-use alternative to monitor motion. Modern computational methods, including deep learning[12], along with large publicly available datasets[13] have enabled pose estimation algorithms, such as OpenPose[14], to produce estimates of body pose from standard video across varying lighting, activity, age, skin color, and angle-of-view[15]. Human pose estimation software, including OpenPose, outputs estimates of the two-dimensional (2D) image-plane positions of joints (e.g., ankles and knees) and other anatomical locations (e.g., heels and pelvis) in each frame of a video (Fig. 1b). These estimates of 2D planar projections are too noisy and biased, due to manually annotated ground truth and planar projection errors, to be used directly for extracting clinically meaningful information such as three-dimensional (3D) gait metrics or treatment indications[16]. Investigators recently predicted cadence

from 2D planar projections[17], but their study included a population of only two impaired subjects and required carefully engineered features, limiting generalizability. Moreover, for predictions that are not directly explained by physical phenomena, such as clinical decisions, feature engineering is particularly difficult. To overcome these limitations, we used deep neural networks (machine learning models that employ multiple artificial neural network layers to learn complex, and potentially nonlinear, relationships between inputs and outputs), which have been shown to be an effective tool for making robust predictions in an impaired population compared with methods using hand-engineered features[18]. Our method capitalizes on 2D pose estimates from video to predict (i) quantitative gait metrics commonly used in clinical gait analysis, and (ii) clinical decisions.

We designed machine learning models to predict clinical gait metrics from trajectories of 2D body poses extracted from videos using OpenPose (Fig. 1b and Supplementary Movie 1). Our models were trained on 1792 videos of 1026 unique patients with cerebral palsy. These videos, along with gold-standard optical motion capture data, were collected as part of a clinical gait analysis. Measures derived from the optical motion capture data served as ground-truth labels for each visit (see Methods). We predicted visit-level gait metrics (i.e., values averaged over multiple strides from multiple experimental trials), since the videos and gold-standard optical motion capture were collected contemporaneously but not simultaneously. These visit-level estimates of values, such as average speed or cadence, are widely adopted in clinical practice. We tested convolutional neural network (CNN), random forest (RF), and ridge regression (RR) models, with the same fixed set of input signals for each model. In the CNN models, we input raw time series; in the other two models (which are not designed for time-series input), we input summary statistics such as mean and percentile. We present the

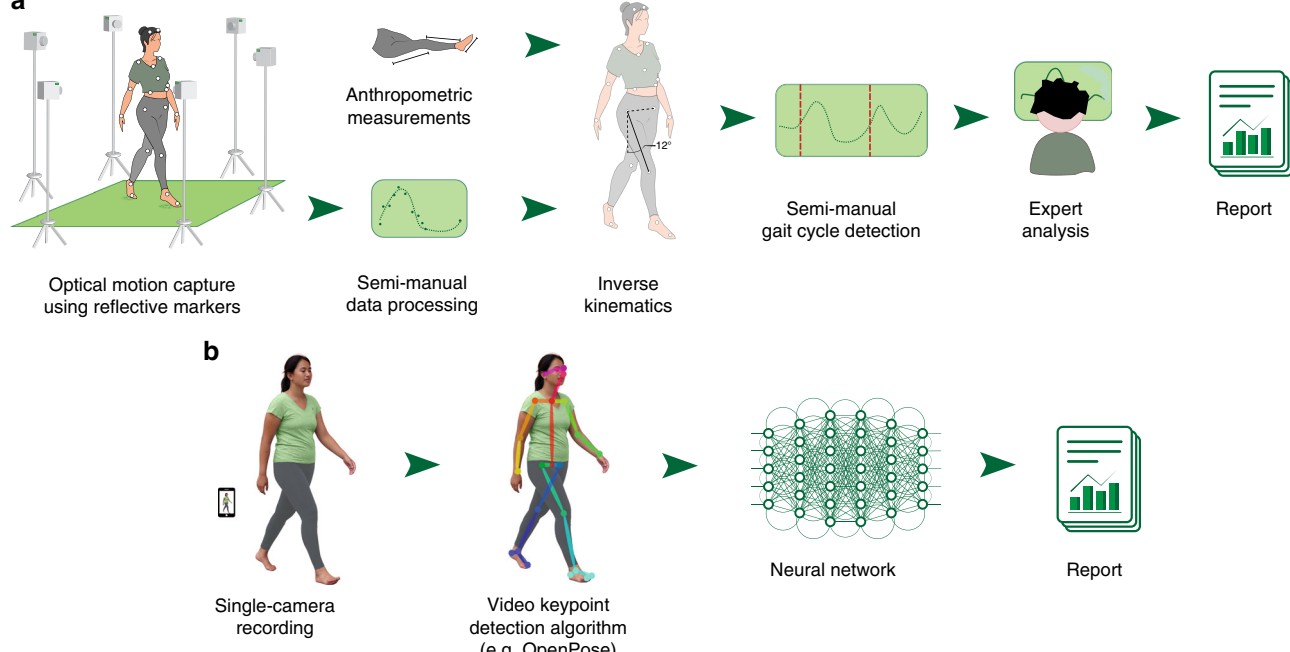

**Fig. 1 Comparison of the current clinical workflow with our video-based workflow. a** In the current clinical workflow, a physical therapist first takes a number of anthropometric measurements and places reflective markers on the patient's body. Several specialized cameras track the positions of these markers, which are later reconstructed into 3D position time series. These signals are converted to joint angles as a function of time and are subsequently processed with algorithms and tools unique to each clinic or laboratory. **b** In our proposed workflow, data are collected using a single commodity camera. We use the OpenPose[14] algorithm to extract trajectories of keypoints from a sagittal-plane video. We present an example input frame, and then the same frame with detected keypoints overlaid. To illustrate the detected pose, the keypoints are connected. Next, these signals are fed into a neural network that extracts clinically relevant metrics. Note that this workflow does not require manual data processing or specialized hardware, allowing monitoring at home.

CNN results since in all cases, the CNN performed as well or better than the other models (Fig. 2); however, more thorough feature engineering specific to each prediction task could improve performance for all model types. Our models, trajectories of anatomic keypoints derived using OpenPose, and ground-truth labels are freely shared at http://github.com/stanfordnmbl/mobile-gaitlab/.

## Results

**Predicting common gait metrics.** We first sought to determine visit-level average walking speed, cadence, and knee flexion angle at maximum extension from a 15 s sagittal-plane walking video. These gait metrics are routinely used as part of diagnostics and treatment planning for cerebral palsy[4] and many other disorders, including Parkinson's disease[19,20], Alzheimer's disease[21,22], osteoarthritis[2,23], stroke[3,24], non-Alzheimer's dementia[25], multiple sclerosis[5,26], and muscular dystrophy[6]. The walking speed, cadence, and knee flexion at maximum extension predicted from

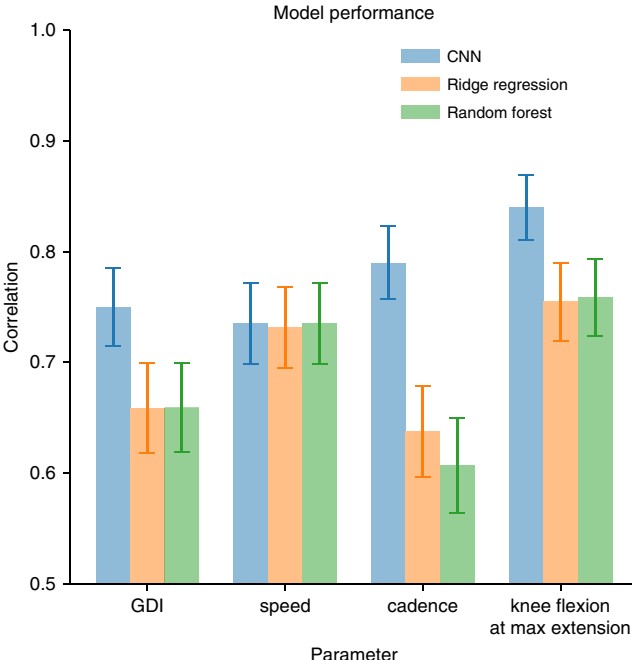

Model performance

video by our best models had correlations of 0.73, 0.79, and, 0.83, respectively, with the ground-truth motion capture data (Table 1 and Fig. 3a–c).

Our model's predictive performance for walking speed was close to the theoretical upper bound given intra-patient stride-to-stride variability. Variability of gait metrics can be decomposed into inter-patient and intra-patient (stride-to-stride) variability[27]. The correlation between our model and ground-truth walking speed was 0.73; thus, our model explained 53% of the observed variance. In the cerebral palsy population, intra-patient stride-to-stride variability in walking speed typically accounts for about 25% of the observed variance in walking speed[28]. Therefore, we do not expect the variance explained to exceed 75% because our video and ground-truth motion capture data were not collected simultaneously, making it infeasible to capture stride-to-stride variability. The remaining 22% of variability likely represented some additional trial-to-trial variability, along with inter-patient variability that the model failed to capture.

Our predictions of knee flexion angle at maximum extension within the gait cycle, a key biomechanical parameter in clinical decision-making, had a correlation of 0.83 with the corresponding ground-truth motion capture data (Fig. 3c). For comparison, the knee flexion angle at maximum extension directly computed from the thigh and shank vectors defined by the hip, knee, and ankle keypoints of OpenPose had a correlation of only 0.51 with the ground-truth value, possibly due in part to the fixed position of the camera and associated projection errors. This implies that information contained in other variables used by our model had substantial predictive power.

**Predicting comprehensive clinical gait measures.** Next, we built models to determine comprehensive clinical measures of motor performance, namely the Gait Deviation Index (GDI)[29] and the Gross Motor Function Classification System (GMFCS) score[30], a measure of self-initiated movement with emphasis on sitting, transfers, and mobility. These metrics are routinely used in clinics to plan treatment and track progression of disorders. Assessing GDI requires full time-series data of 3D joint kinematics measured with motion capture and a biomechanical model, and assessing GMFCS requires trained and experienced medical personnel. To predict GDI and GMFCS from videos, we used the same training algorithms and machine learning model structure that we used for predicting speed, cadence, and knee flexion angle (see Methods).

The accuracies of our GDI and GMFCS predictions were close to the theoretical upper bound given previously reported variability for these measures, indicating that our video analysis could be used as a quantitative assessment of gait outside of a clinic. We predicted visit-level GDI with correlation 0.75 (Fig. 3d), while the intraclass correlation coefficient for visits of the same patient is reported to be 0.81 (0.73–0.89, 95% confidence interval)[31] in children with cerebral palsy (see Methods). Despite the fact that GDI is derived from 3D joint angles, correlations between

**Fig. 2 Comparison of prediction accuracy for models using video signals.** We compare three methods: convolutional neural network (CNN), random forest, and ridge regression. To predict each of the four gait metrics (speed, cadence, GDI, and knee flexion angle at maximum extension), we trained a model on a training set, choosing the best parameters on the validation set. The reported values of bars are the correlation coefficients between the true and predicted values for each metric, evaluated on the test set. Error bars represent standard errors derived using bootstrapping ($n = 200$ bootstrapping trials).

| **Table 1 Model accuracy in predicting continuous visit-level parameters.** | | | |
|---|---|---|---|
| | **True vs. predicted correlation (95% CI)** | **Mean bias (95% CI; p value)** | **Mean absolute error** |
| Walking speed (m/s) | 0.73 (0.66–0.79) | 0.00 (−0.02–0.02; 0.93) | 0.13 |
| Cadence (strides/s) | 0.79 (0.73–0.84) | 0.01 (0.00–0.02; 0.10) | 0.08 |
| Knee flexion (degrees) | 0.83 (0.78–0.87) | 0.33 (−0.40–1.06; 0.38) | 4.8 |
| Gait Deviation Index | 0.75 (0.68–0.81) | 0.54 (−0.33–1.42; 0.22) | 6.5 |

We measured performance of the CNN model for four walking parameters: walking speed, cadence, knee flexion at maximum extension, and Gait Deviation Index (GDI). All statistics were derived from predictions on the test set, i.e., visits that the model has never seen. Bias was computed by subtracting predicted value from observed value. Correlations are reported with 95% confidence interval (CI). All predictions had correlations with true values above 0.73. For perspective, stride-to-stride correlation for GDI is reported to be 0.73–0.89[31], which is comparable with our estimator. We used a two-sided t-test to check if predictions were biased. In each case there was no statistical evidence for rejecting the null hypothesis (no bias).

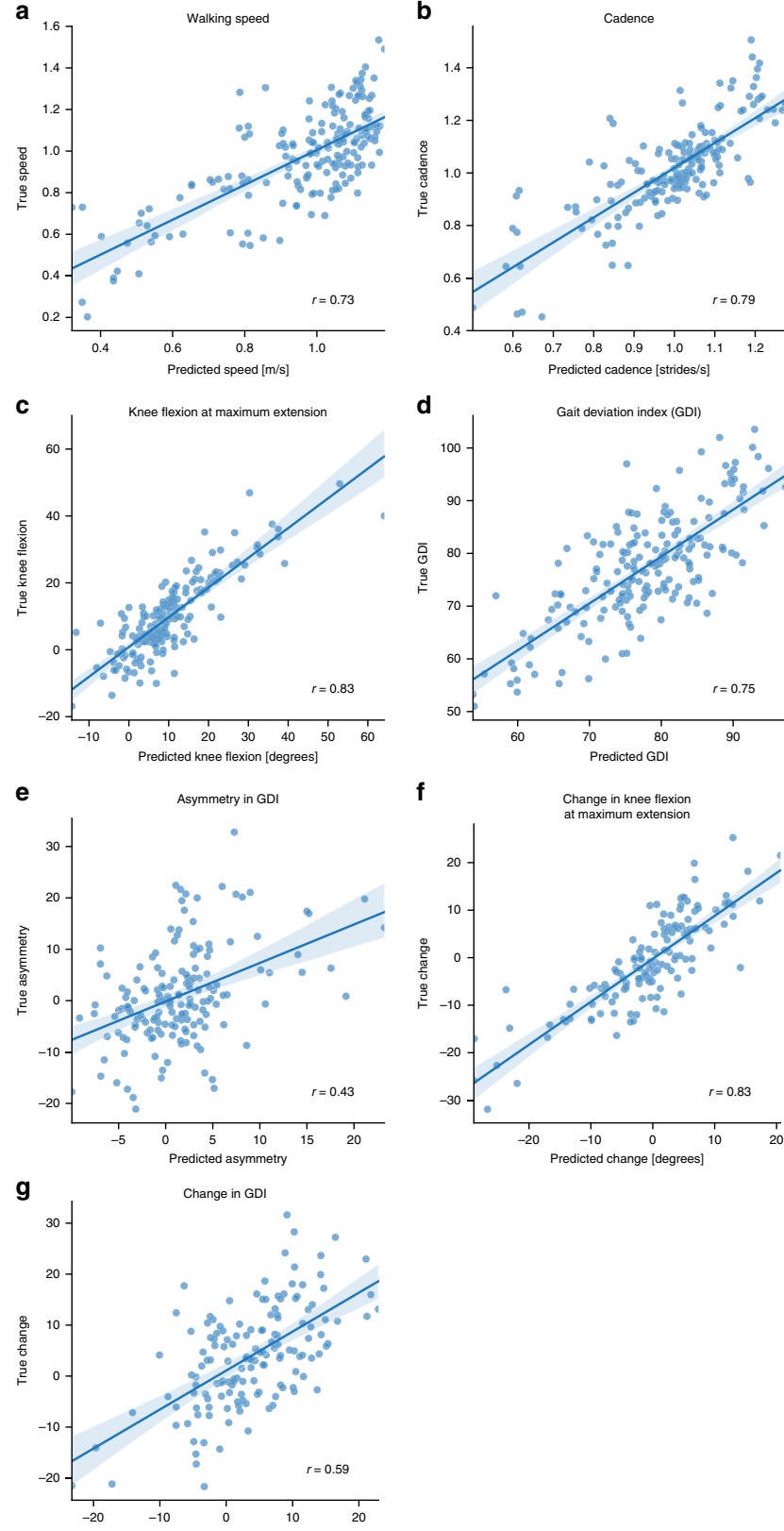

**Fig. 3 Convolution neural network (CNN) model performance.** We evaluated the correlation, *r*, between the true gait metric values from motion capture data and the predicted values from the video keypoint time-series data and our model. Our model predicted (**a**) speed, (**b**) cadence, (**c**) knee flexion angle at maximum extension, and (**d**) Gait Deviation Index. We also did a post-hoc analysis to predict (**e**) asymmetry in GDI, as well as longitudinal changes in (**f**) knee flexion angle at maximum extension and (**g**) GDI. In all plots, the straight blue line corresponds to the best linear fit to predicted vs. observed data while light bands correspond to the 95% confidence interval for the regression curve derived using bootstrapping (*n* = 200 bootstrapping trials).

these joint angles enabled us to predict GDI with high accuracy from 2D video. We predicted GMFCS with weighted kappa of 0.71 (Table 2); inter-rater variability of GMFCS is reported to be 0.76–0.81[32], and agreement between a physician and a parent is 0.48–0.67[33]. The predicted GMFCS scores were correct 66% of the time and always within 1 of the true score. The largest rate of misclassifications occurred while differentiating between GMFCS levels I and II, but this is unsurprising as more information than can be gleaned from a simple 10 m walking task (e.g., about the patient's mobility over a wider range of tasks, terrain, and time) is typically needed to distinguish between these two levels.

We reasoned that remaining unexplained variability in GDI may be due to unobserved information from the frontal and transverse planes. To test this, we computed correlations between the GDI prediction model's residuals and parameters that are not captured by OpenPose from the sagittal view. We found that the residuals between true and predicted GDI were correlated with the patient's mean foot progression angle ($p < 10^{-4}$) and mean hip adduction during gait ($p < 10^{-4}$) as measured by optical

motion capture (Fig. 4). This, along with the higher correlation observed for predicting sagittal-plane knee kinematics, suggests that GDI estimation could be improved with additional views of the patient's gait.

**Predicting longitudinal gait changes and surgical events.** A post-hoc analysis using the predicted gait metrics from single gait visits showed that we partially captured gait asymmetry and longitudinal changes for individual patients. Gait asymmetry may arise from impairments in motor control, asymmetric orthopedic deformity, and asymmetric pain, and can be used to inform clinical decisions[34]. Longitudinal changes can inform clinicians about progression of symptoms and long-term benefits of treatment, since the lack of longitudinal data makes analysis of long-term effects of treatment difficult[35]. We used predicted values from the models described earlier to estimate asymmetry and longitudinal changes, and thus did not train new models for this task. Our predicted gait asymmetry, specifically, the difference in GDI between the two limbs, correlated with the true asymmetry with $r = 0.43$ (Fig. 3e); this lower correlation is expected because we estimate asymmetry as a difference between two noisy predictions of GDI for the left and right limbs. We predicted longitudinal change assuming the true baselines measured in the clinic are known and future values are to be estimated. This framework approximates the use of videos to monitor patients at home after an initial in-clinic gait analysis. The change in knee flexion at maximum extension angle correlated with the true change with $r = 0.83$ (Fig. 3f), while the change in GDI over time correlated with $r = 0.59$ (Fig. 3g). In the case where we did not use baseline GDI in the model, correlations between the difference in model-predicted values and difference in ground-truth clinic-measured values were 0.68 for knee flexion at maximum extension and 0.40 for GDI.

Finally, we sought to predict whether a patient would have surgery in the future, since accurate prediction of treatment might

**Table 2 Model accuracy in predicting the Gross Motor Function Classification System (GMFCS) score.**

|  | True I | True II | True III | True IV |
|---|---|---|---|---|
| Predicted I | 50 | 21 | 0 | 0 |
| Predicted II | 26 | 47 | 1 | 0 |
| Predicted III | 0 | 8 | 22 | 4 |
| Predicted IV | 0 | 0 | 1 | 0 |

The GMFCS score is derived from an expert clinical rater assessing walking, sitting, and use of assistive devices for mobility. The confusion matrix presents our GMFCS prediction based solely on videos in the test set. Prediction using our CNN model has Cohen's kappa = 0.71, which is close to the intra-rater variability in GMFCS. In addition, misclassifications were exclusively by only one level (e.g., True I never predicted to be III or IV).

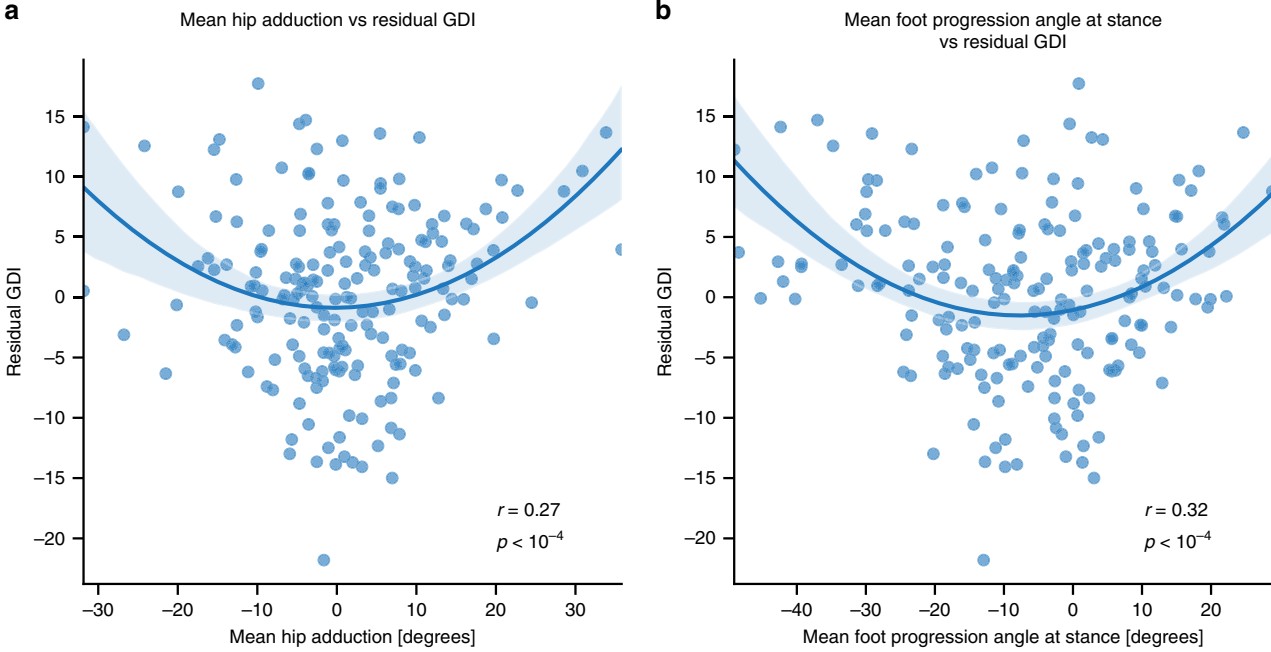

**Fig. 4 Correlation between GDI prediction residuals and non-sagittal-plane kinematics.** The residuals from predicting GDI from video are correlated with the mean (**a**) foot progression and (**b**) hip adduction angles derived from optical motion capture. These correlations suggest that the foot progression and hip adduction angles, which are inputs to the calculation of ground-truth GDI, are not fully captured in the sagittal-plane video. We tried linear and quadratic models and chose the better one by the Bayesian Information Criterion. In each plot, the blue curve corresponds to the best quadratic fit to predicted vs. observed data while the light band corresponds to the 95% confidence interval for the regression curve derived using bootstrapping ($n = 200$ bootstrapping trials). We tested if each fit is significant by using the F-test and we reported corresponding p values.

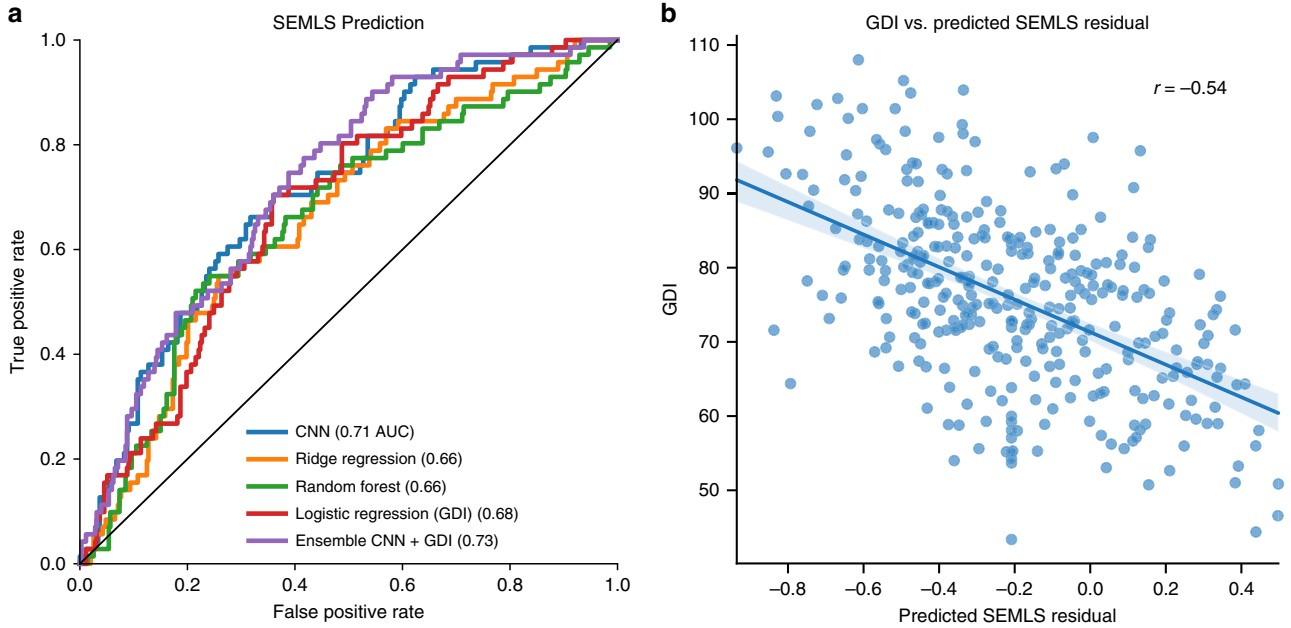

**Fig. 5 Analysis of models for treatment decision prediction. a** Our CNN model outperformed ridge regression and random forest models that used summary statistics of the time series (see Methods) and the logistic regression model using only GDI. **b** Residuals from the CNN model to predict SEMLS treatment decisions correlate with GDI. The straight blue line corresponds to the best linear fit to predicted vs. observed data while the light band corresponds to the 95% confidence interval for the regression curve derived using bootstrapping ($n = 200$ bootstrapping trials).

enable remote screenings in locations with limited access to specialty healthcare. We predicted treatment decisions—specifically, whether a patient received a single-event multilevel surgery (SEMLS) following the analyzed clinical gait visit. This analysis revealed that patient videos contain information that is distinct from GDI and predictive of SEMLS decisions. Our model predicted whether a patient received a SEMLS with Area Under the Receiver Operating Characteristics Curve (AUC) of 0.71 (Fig. 5a). The CNN model slightly outperformed a logistic regression model based on GDI from motion capture (AUC 0.68). An ensemble of our CNN model and the GDI logistic regression model-predicted SEMLS with AUC 0.73, suggesting there is some additional information in GDI compared with our CNN model. We found that residuals of the SEMLS prediction from our CNN model were correlated with GDI with $r = 0.51$ (Fig. 5b), further validating that the two signals have some uncorrelated predictive information.

## Discussion

Our models can help parents and clinicians assess early symptoms of neurological disorders and enable low-cost surveillance of disease progression. For example, GMFCS predictions from our model had better agreement with clinicians' assessments than did parents' assessments. Our methods are dramatically lower in cost than optical motion capture and do not require specialized equipment or training. A therapist or technician need not place markers on a patient, and our models allow the use of commodity hardware (i.e., a single video camera). In our experiments, we downsampled the videos to $640 \times 480$ resolution, a resolution available in most modern mobile phone cameras. In fact, the most recent smartphones are equipped with cameras that record videos in $3840 \times 2160$ resolution at 60 frames per second.

For a robust, production-ready deployment of our models or to extend our models to other patient populations, practitioners would have to address several limitations of our study. First, to use our current models to assess the same set of gait parameters

in children with cerebral palsy, the protocol used in the clinic must be closely followed, including similar camera angles and subject clothing. For deployment under more lax collection protocols, the methods should be tested with new videos recorded by naive users. Second, our study only used sagittal-plane video, making it difficult to capture signals visible mainly in other planes, such as step width. A similar framework to the one we describe in this study could be used to build models that incorporate videos from multiple planes. Third, since videos and motion capture data were collected separately, we could only design our models to capture visit-level parameters. For some applications, stride-wise parameters might be required. With additional data, researchers could test whether our models are suitable for this stride-level prediction, or, if needed, could train new models using a similar framework. In this study, we had access to a large dataset to train our CNN model; if extending our approach to a task where more limited data are available, more extensive feature engineering and classical machine learning models might lead to better results. Finally, the dataset we used was from a single clinical center, and the robustness of our models should be tested with data from other centers. For example, clinical decisions on SEMLS are subjective and must be interpreted in the context of the clinic in which the data was acquired.

Our approach shows the potential for using of video-based pose estimation to predict gait metrics, which could enable community-based measurement and fast and easy quantitative motion analysis of patients in their natural environment. We demonstrated the workflow on children with cerebral palsy and a specific set of gait metrics, but the same method can be applied to any patient population and metric (e.g., step width, maximum hip flexion, and metabolic expenditure). Cost-efficient measurements outside of the clinic can complement and improve clinical practice, enabling clinicians to remotely track rehabilitation or post-surgery outcome and researchers to conduct epidemiological scale clinical studies. This is a significant leap forward from controlled

laboratory tests and allows virtually limitless repeated measures and longitudinal tracking.

## Methods

We analyzed clinical gait analysis videos from patients seen at Gillette Children's Specialty Healthcare. For each video, we used OpenPose[14] to extract time series of anatomical landmarks. Next, we preprocessed these time series to create features for supervised machine learning models. We trained CNN, RF, and RR models to predict gait parameters and clinical decisions, and evaluated model performance on a held-out test set.

**Dataset.** We analyzed a dataset of 1792 videos of 1026 unique patients diagnosed with cerebral palsy seen for a clinical gait analysis at Gillette Children's Specialty Healthcare between 1994 and 2015. Average patient age was 11 years (standard deviation, 5.9). Average height and mass were 133 cm (s.d., 22) and 34 kg (s.d., 17), respectively. About half (473) of these patients had multiple gait visits, allowing us to assess the ability of our models to detect longitudinal changes in gait.

For each patient, optical motion capture (Vicon Motion Systems[36]) data were collected to measure 3D lower extremity joint kinematics and compute gait metrics[37]. These motion capture data were used as ground-truth training labels and were collected at the same visit as the videos, though not simultaneously. While the video system in the gait analysis laboratory has changed multiple times, our post-hoc analysis showed no statistical evidence that these changes affected predictions of our models.

Ground-truth metrics of walking speed, cadence, knee flexion angle at maximum extension, and GDI were computed from optical motion capture data following standard biomechanics practices[38,39]. The data collection protocol at Gillette Children's Specialty Healthcare is described in detail by Schwartz et al.[40]. Briefly, physical therapists placed reflective markers on patients' anatomical landmarks. Specialized, high-frequency cameras and motion capture software tracked the 3D positions of these markers as patients walked over ground. Engineers semi-manually postprocessed these data to fill missing marker measurements, segment data by gait cycle, and compute 3D joint kinematics. These processed data were used to compute gait metrics of interest—specifically, speed, cadence, knee flexion angle at maximum extension, and GDI—per patient and per limb.

The GMFCS score was rated by a physical therapist, based on the observation of the child's function and an interview with the child's parents or guardians. For some visits, surgical recommendations were also recorded.

Videos were collected during the same lab visit as ground-truth motion capture labels, but during a separate walking session without markers. The same protocol was used; i.e., the patient was asked to walk back and forth along a 10 m path 3–5 times. The patient was recorded with a camera ~3–4 m from the line of walking of the patient. The camera was operated by an engineer who rotated it along its vertical axis to follow the patient. Subjects were asked to wear minimal comfortable clothing.

Raw videos in MP4 format with Advanced Video Coding encoding[41] were collected at a resolution of 1280 × 960 and frame rate of 29.97 frames per second. We downsampled videos to 640 × 480, imitating lower-end commodity cameras and matching the resolution of the training data of OpenPose. For each trial we had 500 frames, corresponding to around 16 s of walking.

The study was approved by the University of Minnesota Institutional Review Board (IRB). Patients, and guardians, where appropriate, gave informed written consent at the clinical visit for their data to be included. In accordance with IRB guidelines, all patient data were de-identified prior to any analysis.

**Extracting keypoints with OpenPose.** For each frame in a video, OpenPose returned 2D image-plane coordinates of 25 keypoints together with prediction confidence of each point for each detected person. Reported points were the estimated $(x, y)$ coordinates, in pixels, of the centers of the torso, nose, and pelvis, and centers of the left and right shoulders, elbows, hands, hips, knees, ankles, heels, first and fifth toes, ears, and eyes. Note that OpenPose explicitly distinguished right and left keypoints.

We only analyzed videos with one person visible. After excluding 1443 cases where OpenPose failed to detect patients or where more than one person was visible, the dataset included 1792 videos of 1026 patients. For each video, we worked with a 25-dimensional time series of keypoints across all frames. We centered each univariate time series by subtracting the coordinates of the right hip and scaled all values by dividing by the Euclidean distance between the right hip and the right shoulder. We then smoothed the time series using a one-dimensional unit-variance Gaussian filter. Since some of the downstream machine learning algorithms do not accept missing data, we imputed missing observations using linear interpolation.

For the clinical metrics where values for the right and left limb were computed separately (GDI, knee flexion angle at maximum extension, and SEMLS), we used the time series of keypoints (knee, ankle, heel, and first toe) of the given limb as predictors. Other derived time series, such as the difference in $x$ position between the ipsilateral and contralateral ankle, or joint angles (for knee and ankle), were also computed separately for each limb. We ensured that the training, validation, and test sets contained datapoints coming from different patients. For clinical metrics that were independent of side (speed, cadence, GMFCS), we trained using keypoints from both limbs along with side-independent keypoints and each trial was a single datapoint.

Patients walked back and forth starting with the camera facing their right side. For consistency, and to simplify training, we mirrored the frames and the labels when the patient reversed their walking direction and we kept track of this orientation. As a result, all the walking was aligned so that the camera was always pointing at the right side or a mirrored version of the left side.

**Hand-engineered time series.** We found two derived time series helpful for improving the performance of the neural network model. The first time series was the difference between the $x$-coordinates (horizontal image-plane coordinates) of the left and right ankles throughout time, which approximated the 3D distance between ankle centers. The second time series was the image-plane angle formed by the ankle, knee, and hip keypoints. Specifically, we computed the angle between the vector from the knee to the hip and the vector from the knee to the ankle. This value approximated the true knee flexion angle.

**Architecture and training of CNNs.** CNNs are a type of neural network that use parameter sharing and sparse connectivity to constrain the model architecture and reduce the number of parameters that need to be learned[12]. In our case, the CNN model is a parameterized mapping from a fixed-length time-series data (i.e., anatomical keypoints) to an outcome metric (e.g., speed). The key advantage of CNNs over classical machine learning models was the ability to build accurate models without extensive feature engineering.

The key building block of our model was a 1-D convolutional layer. The input to a 1-D convolutional layer consisted of a $T \times D$ set of neurons, where $T$ was the number of points in the time dimension and $D$ was the depth (in our case, the dimension of the multivariate time-series input into the model). Each 1-D convolutional layer learned the weights of a set of filters of a given length. For instance, suppose we chose to learn filters of length $F$ in our convolutional layer. Each filter connected only the neurons in a local region of time (but extending through the entire depth) to a given neuron in the output layer. Thus, each filter consisted of $FD + 1$ weights (we included the bias term here), so the total number of parameters to an output layer of depth $D_2$ was $(FD + 1)D_2$. Our model architecture is illustrated in Fig. 6.

Each convolutional layer had 32 filters and a filter length of eight. We used the rectified linear unit (ReLU), defined as $f(x) = \max(0, x)$, as the activation function after each convolutional layer. After ReLU, we applied batch normalization (empirically, we found this to have slightly better performance than applying batch normalization before ReLU). We defined a k-convolution block as $k$ 1D convolution layers followed by a max pooling layer and a dropout layer with rate 0.5 (see Fig. 6). We used a mini batch size of 32 and RMSProp (implemented in keras software; keras.io/optimizers) as the optimizer. We experimented with $k \in \{1, 2, 3\}$-convolution blocks to identify sufficient model complexity to capture higher order relations in the time series. After extensive experimentation, we settled on an architecture with $k = 3$.

After selecting the architecture, we did a random search on a small grid to tune the initial learning rate of RMSProp and the learning rate decay schedule. We also searched over different values of the L2 regularization weight ($\lambda$) to apply to the last four convolutional layers. We applied early stopping to iterations of the random search that had problems converging. The final optimal setting of parameters was an initial learning rate of $10^{-3}$, decaying the learning rate by 20% every 10 epochs, and setting $\lambda = 3.16 \times 10^3$ for the L2 regularization. Regularization (both L2 and dropout) is fundamental for our training procedure since our final CNN model has 47,840 trainable parameters, i.e., at the order of magnitude of the training sample.

Our input volume had dimension 124 × 12. The depth was only 12 because preliminary analysis indicated that dropping several of the time series improved performance. We used the same set of features for all models to further simplify feature engineering. The features we used were the normalized $(x, y)$ image-plane coordinates of ankles, knees, hips, first (big) toes, projected angles of the ankle and knee flexion, the distance between the first toe and ankle, and the distance between left ankle and right ankle. Our interpretation of this finding was that some time series, such as the $x$-coordinate of the left ankle, were too noisy to be helpful.

We trained the CNN on 124-frame segments from the videos. We augmented the time-series data using a method sometimes referred to as window slicing, which allowed us to generate many training segments from each video. By covering a variety of starting timepoints, this approach also made the model more robust to variations in the initial frame. From each input time series, $X$, with length 500 in the time dimension and an associated clinical metric (e.g., GDI), $y$, we extracted overlapping segments of 124 frames in length, with each segment separated by 31 frames. Thus for a given datapoint $(y, X)$, we constructed the segments $(y, X[:, 0: 124])$, $(y, X[:, 31: 155])$, …, $(y, X[:, 372: 496])$. Note that each video segment was labeled with the same ground-truth clinical metric $(y)$. We also dropped any segments that had more than 25% of their data missing. For a given video $X_j$, we use the notation $X_j^{(i)}$, $j = 1, 2, …, c(i)$ to refer to its derived segments, where $1 \leq c(i) \leq 12$ counts the number of segments that are in the dataset.

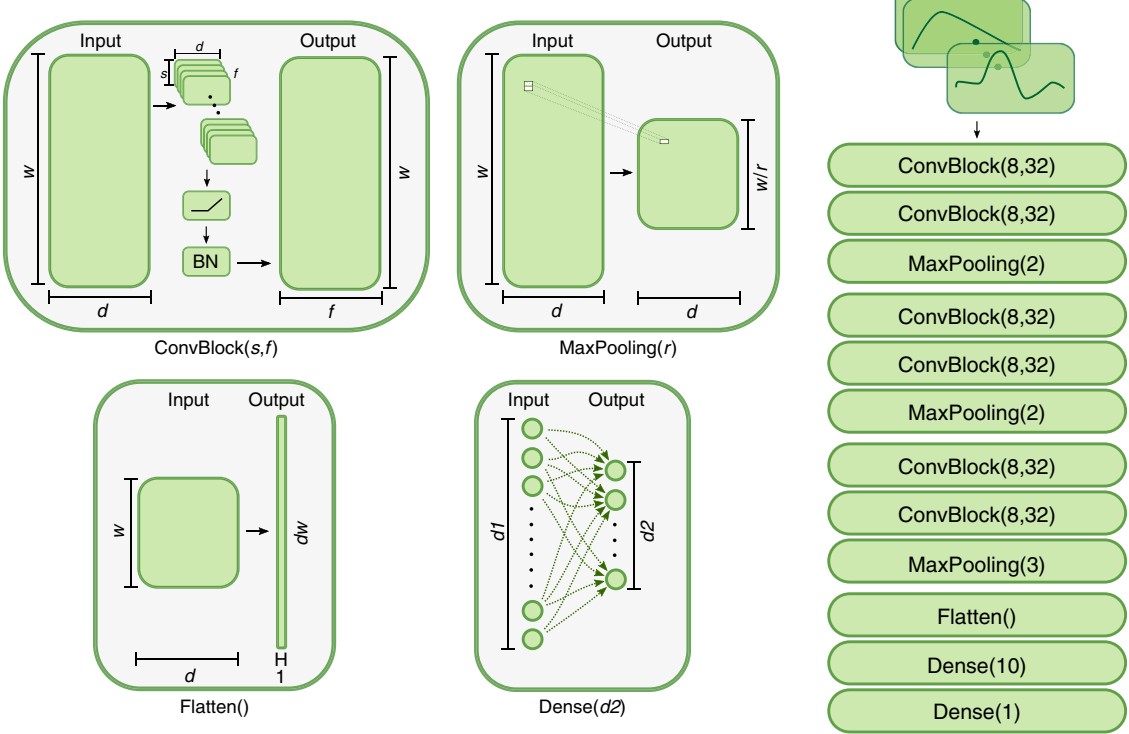

**Fig. 6 Convolutional neural network architecture.** Our CNN is composed of four types of blocks. The convolutional block (ConvBlock) maps a multivariate time series ($w \times d$) into another multivariate time series ($w \times f$) using $f$ parameterized one-dimensional convolutions ($d \times s$), i.e. sliding filters with learnable parameters. Convolutions are followed by a nonlinear activation function and a normalization component. The maximum pooling block (MaxPooling) extracts the maximum value from a sequence of $r$ values, thus reducing the dimensionality from $w$ to $w/r$. The flattening block (Flatten) changes the shape of an array of dimensions $w \times d$ to a vector of dimensions $dw$. Dense block (dense) is a multiple linear regression from $d1$ dimensional space to $d2$ dimensional space with a nonlinear function at the output (see Methods). The diagram on the right shows the sequential combination of these blocks used in our final model.

To train the neural network models we used two loss functions: mean squared error (for regression tasks) or cross-entropy (for classification tasks). The mean squared error is the average squared difference between predicted and true labels. The cross-entropy loss, $L(y, p)$, is a distance between the true and predicted distribution defined as

$$L(y, p) = -(y \log(p) + (1 - y) \log(1 - p)), \tag{1}$$

where $y$ is a true label and $p$ is a predicted probability.

Since some videos had more segments in the training set than others (due to different amounts of missing data), we slightly modified the mean squared error loss function, $MSE\prime(y_i, \hat{y}_i)$, so that videos with more available segments were not overly emphasized during training:

$$MSE\prime(y_i, \hat{y}_i) = (y_i - \hat{y}_i)^2 / c(i), \tag{2}$$

where $y_i$ is a true label, $\hat{y}_i$ is a predicted label, and $c(i)$ is the number of segments available for the $i$-th video.

To get the final predicted gait metric for a given video, we averaged the predicted values from the video segments. However, this averaging operation introduced some bias towards video segments that appeared more often in training (e.g., those in the middle of the video). We reduced this bias by fitting a linear model on the training set, regressing true target values on predicted values. We then used this same linear model to remove the bias of the validation set predictions.

**Ridge regression and random forest**. We compared our deep learning model with classical supervised learning models, including RR and RF. We chose to use RR for its simplicity and its accompanying tools for interpretability and inference, and RF for its robustness in covering nonlinear effects. Both RF and RR require vectors of fixed length as input. The typical way to use these models in the context of time-series data is to first extract high level characteristics of the time series, then use them as features. In our work, we chose to compute the 10th, 25th, 50th, 75th, and 90th percentiles, and the standard deviation of each of 12 univariate time series used in CNNs. Note that for these methods, we used the entire 500-frame multivariate time series from each video rather than 124-frame segments as in the CNNs.

RR is an example of penalized regression that combines L2 regularization with ordinary least squares. It seeks to find weights $\beta$ that minimize the cost function:

$$\sum_{i=1}^{m} (y_i - x_i^T \beta)^2 + \alpha \sum_{j=1}^{p} \beta_j^2, \tag{3}$$

where $x_i$ are the input features, $y_i$ are the true labels, $m$ is the number of observations, and $p$ is the number of input features.

One benefit of RR is that it allows us to trade-off between variance and bias; lower values of $\alpha$ correspond to less regularization, hence greater variance and less bias. The reverse is true for higher values of $\alpha$.

The RF[42] is a robust generalization of decision trees. A single decision tree consists of a series of branches where a new observation is put through a series of binary decisions (e.g., median ankle position <0.5 or ≥0.5). The leaves of the tree at the end of each sequence of branches contain filtered training observations that are then used to make a prediction on the new observation (e.g., using the mean value of the filtered training observations). The RF is comprised of a set of decision trees; for each decision tree in the forest, the variables used to split at each branch (e.g., median ankle position) are stochastically chosen, and the splitting thresholds (e.g., 0.5) are determined accordingly. To build a forest, the user must select hyperparameters, including the depth (i.e., number of sequential branches) of a single tree $d$ and total number of trees $n$. For inference on a new observation, RF models use the average prediction from all trees. Trees are scale invariant and are often a method of choice by practitioners due to their robustness and ability to capture complex nonlinear relationships between the input features and the label to be predicted[43].

We conducted a grid search to tune hyperparameters for the RR and RF models. Instead of doing k-fold cross validation, we used just one validation set to pick the parameters. This was to keep the results consistent with those of the CNN, which only used one validation set for computational reasons.

We found the best setting for the RF was $n = 200$, $d = 10$, and for the RR $\alpha = 0$. The fact that $\alpha = 0$ worked best for the RR suggests that variance was not the main bottleneck in the RR performance.

**Evaluation**. We split the dataset into training, validation, and test sets, such that the test and validation sets contained 10% of all patients (i.e., 1091 patients in the training set and 136 patients in each of the test and validation sets). We ensured

that each patient's videos were only included in one of the sets. For CNNs, after performing window slicing, we ended up with 16,414, 1943, and 1983 segments in the training, validation, and test sets, respectively.

For the regression tasks, we evaluated the goodness of fit for each model using the correlation between true and predicted values in the test set. For the binary classification task (surgery prediction), we used the Receiver Operating Characteristic (ROC) curve to visualize the results and evaluated model performance using the AUC. The ROC curve characterizes how a classifier's true positive rate varies with the false positive rate, and the AUC is the integral of the ROC curve. For the multiclass classification task (GMFCS), we evaluated model performance using the quadratic-weighted Cohen's $\kappa$ defined as

$$\kappa = 1 - \frac{\sum_{i=1}^{k}\sum_{j=1}^{k} w_{ij} x_{ij}}{\sum_{i=1}^{k}\sum_{j=1}^{k} w_{ij} m_{ij}}, \tag{4}$$

where $w_{ij}$, $x_{ij}$, and $m_{ij}$ were weights, observed, and expected (under the null hypothesis of independence) elements of confusion matrices, and $k$ was the number of classes. Quadratic-weighted Cohen's $\kappa$ measures disagreement between the true label and predicted label, penalizing quadratically large errors. For ordinal data, quadratic-weighted Cohen's $\kappa$ can be interpreted as a discrete version of the normalized mean squared error.

To better understand properties of our predictions we used analysis of variance methodology[44]. We observed that total variability of parameters across subjects and trials can be decomposed to three components: patient variability, visit variability, and remaining trial variability. If we define $SS$ as a sum of squares of differences between true values and predictions, one can show that it follows

$$SS = SS_P + SS_V + SS_T, \tag{5}$$

where $SS_P$ is patient-to-patient sum of squares and $SS_V$ is visit-to-visit variability for each patient and, $SS_T$ is trial-to-trial variability for each visit. To assess performance of the model we compare the SS of our model with the SS of the null model (population mean as a predictor). We refer to the ratio of the two as the unexplained variance (or one minus the ratio as the variance explained).

In our work, we were unable to assess $SS_T$ since videos and ground-truth measurements were collected in different trials. However, for most of the gait parameters of interest $SS_T$ is negligible. In fact, if it was large, it would make lab measurements unreliable and such parameters wouldn't be practically useful.

Our metrics based on analysis of variance ignore bias in predictions, so it was important to explicitly check if predictions were unbiased. To that end, for each model we tested if the mean of residuals is significantly different than 0. Each $p$ value was higher than 0.05, indicating there was no statistical evidence of bias at the significance level 0.05. Given a relatively large number of subjects in our study, this also corresponds to tight confidence intervals for the mean of residuals. This reassures us that the bias term can be neglected in the analysis.

**Reporting summary**. Further information on research design is available in the Nature Research Reporting Summary linked to this article.

## Data availability

Video data used in this study were not publicly available due to restrictions on sharing patient health information. These data were processed by Gillette Specialty Healthcare to a de-identified form using OpenPose software as described in the manuscript. The processed de-identified dataset together with clinical variables used in the paper associated with the processed datapoints, were shared by Gillette Specialty Healthcare and are now publicly available at https://simtk.org/projects/video-gaitlab, https://doi.org/10.18735/j0rz-0k12.

## Code availability

We ran OpenPose on a desktop equipped with an NVIDIA Titan X GPU. All other computing was done on a Google Cloud instance with 8 cores and 16 GB of RAM and did not require GPU acceleration. We used scikit-learn (for training the RR and RF models; scikit-learn.org) and keras (for training the CNN; keras.io). SciPy (scipy.org) was also used for smoothing and imputing the time series. Scripts for training machine learning models, the analysis of the results and code used for generating all figures are available in our GitHub repository http://github.com/stanfordnmbl/mobile-gaitlab/.

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

## Acknowledgements
Our research was supported by the Mobilize Center, a National Institutes of Health Big Data to Knowledge (BD2K) Center of Excellence through Grant U54EB020405, and RESTORE Center, a National Institutes of Health Center through Grant P2CHD10191301.

## Author contributions
Conceptualization: L.K., S.L.D., M.H.S. Methodology: L.K., B.Y., J.L.H., A.R., S.L.D., M.H.S. Data curation: L.K., B.Y., A.R., M.H.S. Analysis: L.K., B.Y., J.L.H. Writing: L.K., B.Y., J.L.H., A.R., S.L.D., M.H.S. Funding acquisition: S.L.D., M.H.S.

## Competing interests
The authors declare no competing interests.
