## [Peer Review File · Nature Communications]

Reviewers' Comments:

Reviewer #1:

Remarks to the Author:

Key Results:

The purpose of this study was to report on a novel methodology comparing key metrics on human walking (speed, cadence, symmetry, variability) from a single camera such as a camcorder or cell phone camera (called a commodity camera in their manuscript) to those obtained from a specialized gait laboratory using a multicamera optoelectronic measurement system. The methodology employs pattern recognition software to identify the position of the body in a frame of the camera view and uses neural networks to automate the process of extracting the key metrics and producing a report. The authors propose that reducing the cost of obtaining these data and increasing access to the technology will increase utilization of quantitative motion analysis for surveillance, research, and treatment planning in conditions with movement abnormalities such as Parkinson's disease, osteoarthritis, stroke, cerebral palsy, multiple sclerosis and muscular dystrophy. The key findings from this study include moderate to strong positive correlations ($r=0.75-0.83$) between the new system using machine learning and the gold standard optoelectronic system. The authors conclude that this technique is feasible, extendable to other metrics and gait conditions, and represents a leap forward from laboratory controlled tests to permit quantitative analysis of human movement in natural settings using basic technology guided by advanced analytics.

Validity:

The authors used cerebral palsy as their demonstration diagnosis and based their analysis on 1,792 videos of 1,026 unique patients with cerebral palsy. They divided their data into training (80%), test (10%), and validation (10%) sets. This process ensures that the methodology developed is not unique to a particular dataset as it is tested and validated in additional datasets. The authors do not list the characteristics of the 1,026 unique patients; therefore the results have unknown generalizability to particular populations. The authors argue that this was a demonstration diagnosis and the results should be replicable across other populations; however, this has not been demonstrated in the current study.

Originality and Significance:

Machine learning methods have been used previously to estimate gait parameters or detect the presence of disease as cited by the authors. These methods often use the results of optoelectronic motion capture systems. Others, such as the study by Xu et al. (2015), have used low cost motion sensors such as the Kinect system, but have reported limited accuracy and reliability, and cannot be implemented in community surveillance activities. In another recent study by Luo et al. (2018), the authors used single camera technology using depth and thermal video with convolutional neural networks to detect the type and time spent in activities such as sitting, standing, walking, sleeping, getting assistance or using a commode in a senior living facility homeroom over the course of 1 month. The current study is the first I am aware of that quantifies human movement using a standard video camera and compares the results favorably with an advanced motion capture system.

The implications of this could be substantial. If motion capture could be moved outside the laboratory into more natural settings, the options for analysis are broad and significant.

Xu, Xu, Raymond W. McGorry, Li-Shan Chou, Jia-hua Lin, and Chien-chi Chang. "Accuracy of the Microsoft Kinect™ for measuring gait parameters during treadmill walking." *Gait & posture* 42, no. 2 (2015): 145-151.

Luo, Zelun, Jun-Ting Hsieh, Niranjan Balachandar, Serena Yeung, Guido Pusioli, Jay Luxenberg, Grace Li et al. "Computer vision-based descriptive analytics of seniors' daily activities for long-term health monitoring." *Machine Learning for Healthcare (MLHC)* 2 (2018).

Data and Methodology:

The source data of the 1,792 videos seems appropriate for the study. The data were down sampled to a lower quality employed by current generation cell phone cameras. The methodology has been used for other similar applications, and the authors have openly shared the algorithms used.

Appropriate Use of Statistics:

This reviewer has a passing knowledge of machine learning and neural networks, but I am not an expert on either. Based on my limited knowledge during this review, I believe the authors have used these methods appropriately.

Conclusions:

The authors concluded this technology has potential to increase the use of motion capture technology in the community setting. I agree with their conclusion.

Suggested Improvements:

The manuscript contains much jargon that is not common knowledge to many readers of Nature Research. Expected upper bounds, deep learning, convolutional neural networks, knee flexion at maximum extension are phrases that may not be understandable to readers. Some explanation, brief definition, or reference to their meaning may help increase understanding.

There are no sections in the manuscript looking at the limitations of the study or future directions. These would help to ground the study and put the study findings in context.

Clarity and Context:

Overall, the manuscript reads well and has clear logical flow. The addition of the supplementary material was appreciated as this provides much greater depth of analysis possible for people who want to get more information.

Scope of Reviewer Expertise:

I am familiar with conventional 3D gait analysis methodology, human walking, cerebral palsy, measurement of accuracy and reliability, and statistical methods including linear regression. I am not an expert in machine learning or neural networks.

Reviewer #2:

Remarks to the Author:

Thank you for giving me the opportunity to review this paper on a new way of gait analysis. The paper is well written and clear. It comes from a centre of excellence. The interesting part is that it uses only lateral 2D images and an objective mathematical routine without any influence from experts or experienced staff. This principle offers a broad use and overcomes limitations of present gait analysis due to the need for technical equipment and staff. However, its reduction to 2D and sagittal kinematics limits the gained information. It hence does not replace the conventional instrumented gait analysis completely which offers deeper and more thorough computation and data analysis. I see the new method as presented in the paper as a handy method to gain overview. If the authors agree with this view, I suggest adding a sentence to explain the clinical relevance more clearly.

The authors used the large data base of their gait laboratory and included only patients with cerebral palsy. It is a standard to record simple videos together with the instrumented gait analysis in order to get a clinical impression for the case report and to be able to check in case of questionable data. They used these lateral videos for their project and thus had the possibility to compare their results with the data from the gold-standard 3D gait analysis. These videos were collected in a gait laboratory and thus under relatively standardised conditions. They describe that

the camera was rotated to follow the patient during his walking trials. This rotation distorts the viewing angle on angles and distances strictly in the sagittal plane as a geometrical function of the viewing angle. This affects the knee flexion angle which is projected in more extension, and step length (one factor of gait speed) which is shortened. It is not clear to me how they overcame this 2D problem which depends from the rotation angle of the camera and the one of the legs in space. There is no comment on how much rotation is tolerated.

The authors use an available 2D routine to predict body posture (OpenPose), use this routine for the video frames and analyse the data with a complex mathematical routine. They applied four different mathematical approaches which I am not able to judge as they override my knowledge as a clinician. Convolutional neural networks turned out to produce results closest to the gold-standard of instrumented gait analysis with a very good correlation for most parameters which validates the new method for the gait of patients with cerebral palsy. Besides temporospatial parameters they compute maximal knee extension in stance as an important parameter for indicating surgery. Improving knee extension in stance usually requires a multilevel correction. It is thus not surprising that they could also predict the need for single event multilevel surgery (SEMLS). However, it needs to be stated that such indication depends on the individual centre and they should state that their computations predict when SEMLS would have been indicated at their centre. From the 2D analysis, they are further capable to predict the GDI which is computed from several angles of different planes out of gold-standard gait analysis data. Maybe they could add a short comment to this rather surprising result. They further computed the GMFCS level which is a functional classification. The greatest confusion was to distinguish between level I and II which does not surprise as simple walking is not enough challenge.

I see a place for a broad clinical application of the new tool. It is easy to handle and cheap. However, it does not replace the gold-standard gait analysis in case where more information is required as for precise surgical indications. One problem of conventional standard gait analysis is marker positioning. The new tool works without markers and may be the first step to overcome this problem in the future. The authors describe a novel and relevant new method with a good probability for a major clinical impact. How well it works for other conditions with gait disorders needs to be checked.

Reinald Brunner

Reviewer #3:

Remarks to the Author:

General comments

The authors present a method to assess gait and related clinically important parameters based on 2D video. They use the publicly available library OpenPose to derive time series of 2D joint positions. In a subsequent step, they compare CNN, Ridge Regression (RR), and Random Forests (RF) to estimate (gait) parameters based on those 2D time series. The results are evaluated by correlating the parameters of the 2D estimation with parameters obtained by a reference infrared motion capture system. Additionally, predictions are compared with clinical annotations.

The paper is very well written and allows an easy comprehension of all relevant points.

However, I have the following major concerns regarding the study and paper:

1) The comparison of the proposed method against the reference motion capture method is not sound as not the same walking trials are used to compare both systems (4th paragraph, Supplemental information). Similar gait parameters cannot be expected if gait is performed in different walking trials. For a technical validation, I would expect that at least the same walking trial be used (when predicting trial-wise parameters).

2) It remains unclear, why machine learning (especially deep learning) is necessary to obtain gait

related parameters in the post-processing step. As the pose estimation is already available, it should be easier and of more benefit regarding interpretability to model the data biomechanically or to use signal-processing approaches.

3) The comparison of the CNN with the other machine learning models (RR, RF) is doubtful. The authors state, that the CNN is superior, because it does not require extensive feature engineering. However, for RR and RF, only six features are computed (percentiles + standard deviation) – which cannot be regarded as “extensive” feature engineering. As an example, for the estimation of cadence, it would be very helpful to also use frequency features (there are methods estimating cadence from frequency information (e.g. (Fasel et al., 2017) and others). Additionally, I doubt that differences of 0.03 in the area under the curve can be considered as outperformance of the CNN (regarding SEMLS prediction).

4) In general, why do you think CNNs are superior to other approaches?

I have the following minor concerns:

1) The paper title suggests the analysis of home data. However, the results are obtained from laboratory data. A potential home use case is only mentioned in the outlook. Thus, please remove “Clinical gait analysis at home” from your title.

2) Is it enough to calculate trial-wise gait parameters? Important information may also lie in stride-wise gait parameters. Here, current accuracies regarding speed are below 5 cm/s.

3) Please include information about the accuracy (bias/absolute error) besides correlation.

4) Consider also to use other ways to compare the systems besides correlation (Bland & Altman, 1986).

The general topic of assessing gait using simple technology (e.g., from standard video) is a highly relevant topic. However, the idea of extracting gait parameters from OpenPose output is not new (Sato, Nagashima, Mano, Iwata, & Toda, 2019; Xue et al., 2018). Therefore, I do not see enough innovative height of the proposed method to justify publication in “Nature Communications”. Additionally, as the major concern 1) is not changeable, I recommend rejection of the paper.

Further hints to improve the paper:

Abstract

- Add a short statement about your approach (CNN as post-processing steps and comparison to other ML methods).

- State whether you use stride-wise or trial-wise comparison.

- Add a statement about your clinical population.

- Don't use abbreviations ('max').

Main text

- Check reference 14 – it seems to be incomplete.

- In the text you state a correlation of 0.75 for walking speed, in Fig 2a it is 0.73.

- Please specify somewhere that you calculate trial wise parameters.

- Why do you not expect “the variance explained to exceed 75 %”? Please elaborate.

- As you discuss variance explained for walking speed, also do that for GDI/GMFCS, please.
- Modelling the residuals yields correlations of only 0.18-0.28.
- Which other parameters did you correlate with the residuals?
- Is the result so "significant" that you can argue that it really explains the residual?
- You use lab data where the participants wear few clothes. Please discuss how in real world settings at home, clothing would affect the results obtained by OpenPose.
- You give an outlook regarding step width (which would be harder to assess with sagittal plane cameras). In general, please discuss how the viewing direction would affect the results. I imagine that in home settings, you cannot always guarantee a good viewing direction.

Supplemental information

- How/where did you show that the change of video systems did not change the predictions of the models? Please elaborate.
- Headline "Hand engineered features" -> "Hand engineered time series" (as features are rather the parameters that you input into the ML models).
- How many model parameters in total are estimated in your CNN? How does this number compare to the number of input samples/frames for training? Please discuss.
- For RR and RF, you use 12 univariate time series. From the 25 OpenPose keypoints, which one did you use?

References

Bland, J. M., & Altman, D. G. (1986). Statistical methods for assessing agreement between two methods of clinical measurement. *Lancet*, 1(8476), 307–310. [https://doi.org/10.1016/S0140-6736\(86\)90837-8](https://doi.org/10.1016/S0140-6736(86)90837-8)

Fasel, B., Duc, C., Dadashi, F., Bardyn, F., Savary, M., Farine, P. A., & Aminian, K. (2017). A wrist sensor and algorithm to determine instantaneous walking cadence and speed in daily life walking. *Medical and Biological Engineering and Computing*, 55(10), 1773–1785. <https://doi.org/10.1007/s11517-017-1621-2>

Sato, K., Nagashima, Y., Mano, T., Iwata, A., & Toda, T. (2019). Quantifying normal and parkinsonian gait features from home movies: Practical application of a deep learning-based 2D pose estimator. *PLoS ONE*, 14(11), 1–15. <https://doi.org/10.1371/journal.pone.0223549>

Xue, D., Sayana, A., Darke, E., Shen, K., Hsieh, J.-T., Luo, Z., ... Fei-Fei, L. (2018). Vision-Based Gait Analysis for Senior Care. In *Machine Learning for Health (ML4H) Workshop at NeurIPS 2018*. Retrieved from <http://arxiv.org/abs/1812.00169>

Reviewer #4:

Remarks to the Author:

The authors present a manuscript on the use of OpenPose to track walking in children with cerebral palsy from laboratory based sagittal plane videos. The tracking of key points is then used as input to 3 different models (convolution neural networks, random forest and ridge regression) to predict gait metrics and likelihood of surgery. The manuscript is clear and follows a logical order. The use of OpenPose and neural networks in the field of gait analysis is novel and there is

certainly evidence to suggest that this is an emerging area for the field which will certainly be of interest. The authors have provided enough mythological detail and made their source code available online to allow their experimental setup to be reproduced.

OpenPose was trained on a large varied dataset and is highly likely to be applicable to this patient population. While the authors claim these methods would allow for at home clinical gait analysis on a smart-phone I find this claim overstated. The videos used in this study are from a laboratory setting, with experienced personnel, the camera on a tripod tracking the subject from the side, there is a clean background and the patient in wearing minimal clothing, exposing the body parts of interest. Videos taken in the 'real world' would differ greatly. It would be more reasonable to state that this approach would provide a tool for monitoring progression or managing time of referral to appropriate services.

Further consideration should be given to the reporting of the statistical analysis. In the text, only the correlations of model predicted metrics with ground truth has been reported, no measure of variance has been included, although displayed in the figures. This is relevant especially as the authors are looking to explain the variance in the results.

Prediction of longitudinal change in gait metrics requires further consideration. The correlations found for those that underwent SEMLS is clearly influenced by the point at the beginning of the graph. There is also likely Interactions between terms that need to be taken into consideration. Mismatch of results, graph for walking speed correlation 0.73, text states 0.75.

We thank the reviewers for their thoughtful comments and suggestions. We have made substantial changes to the manuscript and provide additional analysis of our results to accommodate the reviewers' excellent suggestions. These updates have significantly improved the quality and clarity of the manuscript. We provide responses to each reviewer comment below.

Reviewer #1 (Remarks to the Author):

Validity:

The authors used cerebral palsy as their demonstration diagnosis and based their analysis on 1,792 videos of 1,026 unique patients with cerebral palsy. They divided their data into training (80%), test (10%), and validation (10%) sets. This process ensures that the methodology developed is not unique to a particular dataset as it is tested and validated in additional datasets. The authors do not list the characteristics of the 1,026 unique patients; therefore the results have unknown generalizability to particular populations. The authors argue that this was a demonstration diagnosis and the results should be replicable across other populations; however, this has not been demonstrated in the current study.

We agree with the reviewer's comment and removed generalizability claims. The revised manuscript states that further research is needed to test whether our method applies to other populations; see the section on limitations of the study (lines 163-180). We also provide more statistics on the clinical population in our study in Materials & Methods (lines 277-279).

Originality and Significance:

Machine learning methods have been used previously to estimate gait parameters or detect the presence of disease as cited by the authors. These methods often use the results of optoelectronic motion capture systems. Others, such as the study by Xu et al. (2015), have used low cost motion sensors such as the Kinect system, but have reported limited accuracy and reliability, and cannot be implemented in community surveillance activities. In another recent study by Luo et al. (2018), the authors used single camera technology using depth and thermal video with convolutional neural networks to detect the type and time spent in activities such as sitting, standing, walking, sleeping, getting assistance or using a commode in a senior living facility homeroom over the course of 1 month. The current study is the first I am aware of that quantifies human movement using a standard video camera and compares the results favorably with an advanced motion capture system.

The implications of this could be substantial. If motion capture could be moved outside the laboratory into more natural settings, the options for analysis are broad and significant.

We thank the reviewer for appreciating the originality of our research. We acknowledge the growing body of work done in the context of movement analysis from videos and we added the suggested references (Luo et al. 2018; Xu et al. 2015), as well as references mentioned by other reviewers.

Data and Methodology:

The source data of the 1,792 videos seems appropriate for the study. The data were down sampled to a lower quality employed by current generation cell phone cameras. The methodology has been used for other similar applications, and the authors have openly shared the algorithms used.

Appropriate Use of Statistics:

This reviewer has a passing knowledge of machine learning and neural networks, but I am not an expert on either. Based on my limited knowledge during this review, I believe the authors have used these methods appropriately.

Conclusions:

The authors concluded this technology has potential to increase the use of motion capture technology in the community setting. I agree with their conclusion.

We thank the reviewer for acknowledging originality, impact, and potential of our work.

Suggested Improvements:

The manuscript contains much jargon that is not common knowledge to many readers of Nature Research. Expected upper bounds, deep learning, convolutional neural networks, knee flexion at maximum extension are phrases that may not be understandable to readers. Some explanation, brief definition, or reference to their meaning may help increase understanding.

In the updated version of the manuscript we reduced the specialized language from the fields of computer science and biomechanics and defined essential terminology. Changes include the following:

- We used general terms like “computational methods” or “machine learning” rather than “deep learning” where it was appropriate (e.g., lines 35-36). We also add a brief definition of what we mean by deep learning (lines 48-51).
- We changed convolutional neural networks to “machine learning models” wherever the type of the model is not of main importance (e.g., line 54).
- We removed terms like neural network architecture from the main text (lines 101-103) as these technical details are explained in Materials & Methods.
- We updated the text to include more explicit descriptions instead of terms that require some background knowledge (e.g., we changed “2D data” to “2D planar projections” in line 44).

- We added a brief description of the Random Forest method in Materials & Methods (lines 431-439).

There are no sections in the manuscript looking at the limitations of the study or future directions. These would help to ground the study and put the study findings in context.

We thank the reviewer for pointing this out. In the updated version of the manuscript, we state limitations (lines 163-180).

Clarity and Context:

Overall, the manuscript reads well and has clear logical flow. The addition of the supplementary material was appreciated as this provides much greater depth of analysis possible for people who want to get more information.

We thank the reviewer for noting the paper was clear and are glad that the Materials & Methods were helpful.

Scope of Reviewer Expertise:

I am familiar with conventional 3D gait analysis methodology, human walking, cerebral palsy, measurement of accuracy and reliability, and statistical methods including linear regression. I am not an expert in machine learning or neural networks.

Reviewer #2 (Remarks to the Author):

Thank you for giving me the opportunity to review this paper on a new way of gait analysis. The paper is well written and clear. It comes from a centre of excellence. The interesting part is that it uses only lateral 2D images and an objective mathematical routine without any influence from experts or experienced staff. This principle offers a broad use and overcomes limitations of present gait analysis due to the need for technical equipment and staff. However, its reduction to 2D and sagittal kinematics limits the gained information. It hence does not replace the conventional instrumented gait analysis completely which offers deeper and more thorough computation and data analysis. I see the new method as presented in the paper as a handy method to gain overview. If the authors agree with this view, I suggest adding a sentence to explain the clinical relevance more clearly.

Thank you for reviewing our manuscript and for appreciating the potential impact of our approach. We agree that our method should be perceived as a simple means to gain an overview, which would complement the deeper analysis that is possible in a laboratory setting. For example, our approach could be used for screening pre-diagnostics at home or for post-surgical monitoring, when the cost of a full motion capture trial precludes its frequent use. Our method complements current clinical practice by providing a tool that enables cost-efficient tracking of disease progression and surgery outcomes. Additionally, we also see a potential for our method to provide large-scale quantitative epidemiological data on movement-related disorders, which has not been possible so far due to the affordability of motion capture. We added an explanatory sentence in the results (lines 128-130) and in the discussion paragraph (lines 181-182 and 185-188). Moreover, we updated the clinical use case scenario to combine clinical and at-home measurements, showing substantial value of our method as a complement to current practice (lines 134-141).

The authors used the large data base of their gait laboratory and included only patients with cerebral palsy. It is a standard to record simple videos together with the instrumented gait analysis in order to get a clinical impression for the case report and to be able to check in case of questionable data. They used these lateral videos for their project and thus had the possibility to compare their results with the data from the gold-standard 3D gait analysis. These videos were collected in a gait laboratory and thus under relatively standardised conditions. They describe that the camera was rotated to follow the patient during his walking trials. This rotation distorts the viewing angle on angles and distances strictly in the sagittal plane as a geometrical function of the viewing angle. This affects the knee flexion angle which is projected in more extension, and step length (one factor of gait speed) which is shortened. It is not clear to me how they overcame this 2D problem which depends from the rotation angle of the camera and the one of the legs in space. There is no comment on how much rotation is tolerated.

The changing angle of the camera implies that we are operating in a distorted sagittal plane and 2D measurements cannot be interpreted directly. However, our algorithm is composed of two parts: (i) extraction of 2D estimates and (ii) a machine learning model on top of the 2D estimates (Figure 1c). While the first part returns distorted observations, the second part corrects for these errors as long as conditions are similar to those used in our dataset (10 meter walk and a camera 3-4 meters from the subject, rotating to capture the subject — corresponding to angles from around -50 to 50 degrees). The details of video collection are reported in Materials & Methods, subsection Dataset, and now we also directly refer to this in the main text (lines 164-166).

The correction obtained with machine learning can be observed directly. For example, we compared the maximum knee flexion directly obtained from OpenPose, with the estimate obtained from our machine learning model. Correlation between the ground truth and the estimates from OpenPose was 0.51 while with our machine learning model it was 0.83, as we note in the manuscript (lines 88-94). Similarly, for cadence our best model achieved correlation with the gold standard 0.79, while a naïve signal processing-based heuristic had correlation 0.6 (Response Figure 1).

This improvement is directly related to the discussion of the previous point mentioned by the reviewer regarding the limitation of the 2D projection and highlights the essence of our method. The direct projection loses some 3D information due to the distorted sagittal plane and properties of the projection. However, our machine learning models trained on a large dataset were able to recover some of the 3D information.

The authors use an available 2D routine to predict body posture (OpenPose), use this routine for the video frames and analyse the data with a complex mathematical routine. They applied four different mathematical approaches which I am not able to judge as they override my knowledge as a clinician. Convolutional neural networks turned out to produce results closest to the gold-standard of instrumented gait analysis with a very good correlation for most parameters which validates the new method for the gait of patients with cerebral palsy. Besides temporospatial parameters they compute maximal knee extension in stance as an important parameter for indicating surgery. Improving knee extension in stance usually requires a multilevel correction. It is thus not surprising that they could also predict the need for single event multilevel surgery (SEMLS). However, it needs to be stated that such indication depends on the individual centre and they should state that their computations predict when SEMLS would have been indicated at their centre.

We appreciate this comment and added a disclaimer to clarify that clinical decisions are subjective and are influenced by practices and protocols in a given clinic (lines 177-180). Results, particularly for decision support, should be validated in multiple centers, as we agree that practices vary between centers.

From the 2D analysis, they are further capable to predict the GDI which is computed from several angles of different planes out of gold-standard gait analysis data. Maybe they could add a short comment to this rather surprising result. They further computed the GMFCS level which is a functional classification. The greatest confusion was to distinguish between level I and II which does not surprise as simple walking is not enough challenge.

We were also encouraged to find that the 2D videos produced predictions of the GDI that were strongly correlated with gold standard values. We have better described this in the updated manuscript (lines 109-110). We agree that misclassification of GMCS score levels I and II is to be expected and incorporated the reviewer's point on GMFCS in the main text (lines 113-116)

I see a place for a broad clinical application of the new tool. It is easy to handle and cheap. However, it does not replace the gold-standard gait analysis in case where more information is required as for precise surgical indications. One problem of conventional standard gait analysis is marker positioning. The new tool works without markers and may be the first step to overcome this problem in the future. The authors describe a novel and relevant new method with a good probability for a major clinical impact. How well it works for other conditions with gait disorders needs to be checked.

Reinald Brunner

Thank you for appreciating our work and for all the helpful comments. We agree that this method will not replace current gold-standard gait analysis but can be a complementary method due to its affordability. We have made these points more clearly in the revised manuscript. We updated this in the discussion (lines 185-189).

Reviewer #3 (Remarks to the Author):

General comments

The authors present a method to assess gait and related clinically important parameters based on 2D video. They use the publicly available library OpenPose to derive time series of 2D joint positions. In a subsequent step, they compare CNN, Ridge Regression (RR), and Random Forests (RF) to estimate (gait) parameters based on those 2D time series. The results are evaluated by correlating the parameters of the 2D estimation with parameters obtained by a reference infrared motion capture system. Additionally, predictions are compared with clinical annotations.

The paper is very well written and allows an easy comprehension of all relevant points.

However, I have the following major concerns regarding the study and paper:

We thank the reviewer for providing extensive and constructive feedback. Based on the comments from the reviewer, we have performed additional analysis, as described below. We have also implemented changes throughout the manuscript based on this feedback, which has greatly improved the paper.

1) The comparison of the proposed method against the reference motion capture method is not sound as not the same walking trials are used to compare both systems (4th paragraph, Supplemental information). Similar gait parameters cannot be expected if gait is performed in different walking trials. For a technical validation, I would expect that at least the same walking trial be used (when predicting trial-wise parameters).

We agree that when validating a new measurement methodology, data for both the new “instrument” and the gold standard should be collected simultaneously. For trial-wise parameters the new instrument needs to be validated against the gold standard on the same trial, while for visit-level parameters it needs to be validated against gold standard from the same visit. In our work we cover the latter case — measurements are taken during the same visit. We agree with the reviewer that this validation methodology does not allow us to validate trial-wise prediction, but it does allow us to estimate visit-level prediction. In clinical practice for cerebral palsy, most parameters are collected with stride-to-stride granularity, but they are then averaged across strides, and this average is used as a measurement from the visit. This is also true for assessments in other disorders, such as Parkinson’s disease or osteoarthritis. In our work we sought to estimate this type of visit-level measurement. We updated the manuscript to make it clear that we focused on visit-level predictions (lines 57-61).

From a statistical standpoint, trial-to-trial variability is small compared to visit-to-visit and patient-to-patient variability, which we show here. The sum of squares of a measurement from a trial, as a nested model, can be decomposed to

$$SS = SS_P + SS_V + SS_T,$$

where SS is the total sum of squares (or variance if normalized) of per trial measurements across the entire population, SS_P is patient-to-patient sum of squares and SS_V is visit-to-visit variability for each patient and, SS_T is trial-to-trial variability for each visit. Similar decomposition holds true for variances. For GDI in the population of patients with CP, visit-to-visit variability accounts for 81% of the variance (73% - 89%, 95% confidence interval; (Rasmussen et al. 2015). We have added these details in the main text (lines 106-110) and in the Materials & Methods (Section Evaluation).

Our objective was to show that neural networks for pose estimation coupled with appropriate denoising techniques are sufficient for providing clinically meaningful measurements. We agree that our method can be improved by collecting more granular data and making additional adaptations for a production-ready version. We commented on these opportunities and limitations in the revised manuscript (lines 48-51 and 171-177).

2) It remains unclear, why machine learning (especially deep learning) is necessary to obtain gait related parameters in the post-processing step. As the pose estimation is already available, it should be easier and of more benefit regarding interpretability to model the data biomechanically or to use signal-processing approaches.

We agree that the raw outputs of OpenPose can be used to derive gait parameters such as joint angles and speed with the help of biomechanical models and signal processing. In the manuscript, we directly compared measurements with and without the second layer of machine learning models. We found a correlation of 0.51 between the knee flexion at maximum extension computed directly from OpenPose (with the help of a simple biomechanical model) and the gold standard value. These metrics can be markedly improved using machine learning. Our neural network on top of the OpenPose output achieved 0.83 correlation for the same task. OpenPose only gives planar projections, which leads to errors and distortions that our machine learning model can correct.

We also conducted an analysis to compare a signal-processing approach for estimating cadence to our model's prediction. In particular, we estimated cadence as the peak in the periodogram for the left ankle keypoint, similar to what was suggested in the literature for wrist-worn sensors (Fasel et al. 2017) and referenced in the reviewer comment below. We found that this estimator correlates with the gold standard with $r = 0.60$ (Response Figure 1), while our best model predicts cadence with $r =$

0.79. While signal processing results can be further improved it would require extensive analysis for each new parameter.

Response Figure 1. An estimator of cadence based on the peak in the periodogram. Cadence is estimated as the inverted step frequency and it correlates with ground truth with $r=0.6$. Left: Periodogram of one patient with the real cadence indicated by the vertical line. Right: Predicted cadence (x axis) vs. cadence derived from motion capture (y axis).

Together, these comparisons motivate the use of machine learning-based post-processing for denoising the OpenPose output. Numerous papers point to the fact that OpenPose raw outputs are too inaccurate for meaningful clinical use (Seethapathi et al. 2019). Our paper illustrates we can improve these results by denoising the data using machine learning models.

Regarding the complexity of the post-processing model, we agree there is a trade-off between interpretability and robustness of the model. For this reason, we also tried more classical models, including linear regression and random forests as baseline models. These models require less data and are easier to interpret, while still achieving comparable performance. While we carefully designed our baseline models, they are used as an illustration rather than an exhaustive analysis. There is extensive literature on methods that can be used for such multivariate time series analysis, including time series methods, signal processing, Markov models, multivariate statistics, and more. Extensive analysis of these methods is beyond the scope of this work. Here, our key objective was to show that existing data analysis methods enable automatic and clinically-relevant analysis of videos. We updated the manuscript to make these objectives clearer (lines 44-51) and to acknowledge that more

task- and model-specific feature engineering and design could improve performance for all model types (lines 67-68).

3) The comparison of the CNN with the other machine learning models (RR, RF) is doubtful. The authors state, that the CNN is superior, because it does not require extensive feature engineering. However, for RR and RF, only six features are computed (percentiles + standard deviation) – which cannot be regarded as “extensive” feature engineering. As an example, for the estimation of cadence, it would be very helpful to also use frequency features (there are methods estimating cadence from frequency information (e.g. (Fasel et al. 2017) and others). Additionally, I doubt that differences of 0.03 in the area under the curve can be considered as outperformance of the CNN (regarding SEMLS prediction).

We agree with the reviewer’s observations:

1. Our feature engineering for RR and RF is not exhaustive.
2. With more exhaustive feature engineering RR and RF models could potentially outperform CNNs.
3. A difference of 0.03 in AUC should not be considered as out-performance.

However, the CNNs achieved very good performance without any feature engineering and by applying the same network architecture and training methodology for predicting each of the variables — making it useful across different tasks. It is likely that performance of RR and RF can be improved (potentially outperforming CNNs) with additional feature engineering, but this would require considerable effort for each new parameter, and it would be difficult for more complex holistic parameters such as GDI and GMFCS, or clinical decisions. We updated the manuscript to make clear that we favor CNNs for simplicity of application rather than superiority of predictive performance (lines 44-51 and 62-68).

We agree that differences of AUC 0.03 can correspond to 1-3 percent point changes in sensitivity and specificity. These small changes are not relevant in the scope of this study and we changed the wording in the manuscript (see e.g. line 146). All models can be further improved for a production-ready application of our methods, as we state in the revised manuscript (lines 65-68 and 174-177).

4) In general, why do you think CNNs are superior to other approaches?

We do not perceive CNNs as superior to other methods, and we updated the manuscript to reflect that. In most situations, our study included, there are advantages and disadvantages of CNNs.

Advantages of the CNN model in our study include: (i) robustness to the task (we used the same model architecture for all applications), (ii) highest performance compared to reference models across

all parameters of interest, and (iii) no need for extensive feature engineering. However, these advantages come at the cost: we can train these models only with relatively large amounts of data, and it is harder to interpret them compared to classical statistical and signal processing models. These advantages are now stated in (lines 48-51 and 65-68)

Given a very large dataset we can afford using less data-efficient methods (such as CNNs), which allows us to achieve high performance without much feature engineering or model tuning. However, for further analysis of specific tasks and features, more elementary methods may yield more interpretable results or even more accurate estimates, particularly in cases where less data is available. We highlighted this in the limitations section (lines 174-177).

Please note that this answer refers to the CNN models built to predict gait parameters using the post-processed OpenPose outputs. The deep neural network in OpenPose for detecting body keypoints has been proved to be superior to existing classical approaches to pose estimation problems on a Microsoft COCO: Common Objects in Context dataset with over 250,000 people annotated in very diverse images (Lin et al. 2014).

I have the following minor concerns:

1) The paper title suggests the analysis of home data. However, the results are obtained from laboratory data. A potential home use case is only mentioned in the outlook. Thus, please remove “Clinical gait analysis at home” from your title.

We agree with this observation. We changed the title to “Deep neural networks enable quantitative movement analysis using single-camera videos”.

2) Is it enough to calculate trial-wise gait parameters? Important information may also lie in stride-wise gait parameters. Here, current accuracies regarding speed are below 5 cm/s.

In clinical practice for cerebral palsy, measurements are taken stride-wise, but then are commonly either aggregated to a visit-level average and/or one of the trials is used as a representative trial for the visit. An accurate at-home estimate of trial-wise (or visit-level) parameters could be directly used in current practice. We agree that in some applications, stride-wise estimates might be of importance. Our model training and testing methodology generalizes to other granularity (trial-wise or stride-wise) if provided with the appropriate data. We modified the manuscript to highlight these observations (lines 58-62 and 171-174).

3) Please include information about the accuracy (bias/absolute error) besides correlation.

We analyzed bias but did not report it in the first version of the manuscript. In the updated version, we now state that we performed the appropriate analysis and t-tests that revealed no statistical evidence for significant bias, i.e. null hypothesis bias equals zero (Materials & Methods, Evaluation). We agree that a more detailed description of accuracy results is helpful. We computed other statistics and included a new table in the manuscript summarizing these results (Table 1).

4) Consider also to use other ways to compare the systems besides correlation (Bland, Martin Bland, and Altman 1986).

We performed Bland-Altman analysis for predictions on the test set and found very small bias (statistically not significant) and no overrepresentation of outliers (Response Figure 2).

Response Figure 2: Bland-Altman plots for four parameters predicted using our best model. For each metric we see relatively small bias, most of the errors within one standard deviation,

and no major outliers. Slight bias at high speeds and high cadence may indicate that our estimator is conservative, but this finding is not statistically significant.

The general topic of assessing gait using simple technology (e.g., from standard video) is a highly relevant topic. However, the idea of extracting gait parameters from OpenPose output is not new (Sato et al. 2019; Xue et al. 2018). Therefore, I do not see enough innovative height of the proposed method to justify publication in “Nature Communications”. Additionally, as the major concern 1) is not changeable, I recommend rejection of the paper.

We appreciate all of the reviewer’s insightful comments and responding to them has improved the quality of our work. We hope that the revised manuscript establishes value sufficient for changing the reviewer’s opinion of the paper.

First, while we agree that the idea is not new, we address key challenges for existing OpenPose-based approaches. Currently, it is commonly believed that OpenPose output is not sufficiently accurate for downstream analysis (Seethapathi et al. 2019) because of inherent bias in planar projections and bias in the annotations of joints and anatomy compared to clinical and research practice. We show that OpenPose outputs can be denoised using statistical methods and a sufficiently large dataset. This represents a significant conceptual and technical leap and has the potential to fundamentally change paradigms of motion analysis.

Second, while the reviewer’s major concern (1) is not changeable, we believe that in this response letter we provided evidence that while our approach is unable to measure trial-to-trial variability, it provides a clinically-relevant framework for assessing patient-to-patient and visit-to-visit variability. Our results can be interpreted as lower bounds of the actual trial-to-trial performance if more and better data are collected for a production-ready implementation of our methodology.

Third, we believe that the model training framework, trained models, as well as data released with this work (trajectories of body landmarks for 1,792 videos and 1,026 unique patients) can lead to epidemiological-scale gait analysis, particularly for cerebral palsy. Our experience in deploying easy to use libraries, software, and datasets will allow us to engage the community to build on our work and data.

We also updated references and the literature review. We highly value the work done by Soto et al. (2019); however, while their feature engineering approach is well suited for basic gait parameters (e.g. speed and cadence) for which we can derive proxies from keypoint time series, it is difficult to extend it to complex parameters such as GDI, GMFCS, and clinical decisions. Moreover, it requires feature engineering for each new parameter, while our method works without modifications across

multiple gait parameters. Finally, the analysis from Soto was performed on a cohort of 117 healthy subjects and two subjects with Parkinson's disease, which is a limited sample. Signal processing heuristics perform well in such settings, but they tend to fail when applied to impaired populations, and that is where our data-driven methods proved superior. Xue et al., (2018) in their conference workshop paper provide a framework for a proof of concept study. In our work we present more mature analysis on a large cohort in a more controlled setting, leading to stronger conclusions.

Further hints to improve the paper:

Abstract

- **Add a short statement about your approach (CNN as post-processing steps and comparison to other ML methods).**

We added a sentence, as suggested (lines 9-12).

- **State whether you use stride-wise or trial-wise comparison.**

In the updated abstract we now explicitly explain that parameters are calculated and validated at the visit level (lines 14-15)

- **Add a statement about your clinical population.**

We updated the manuscript to describe that we tested the method on children with cerebral palsy, (line 11)

- **Don't use abbreviations ('max').**

We changed max to maximum (line 13)

Main text

- **Check reference 14 – it seems to be incomplete.**

We added a link to the preprint (following guidelines on citing preprints). In the updated version of the manuscript it is reference number 16.

- **In the text you state a correlation of 0.75 for walking speed, in Fig 2a it is 0.73.**

We changed the text to 0.73, which is the correct value (lines 12, 76, 81). We thank the reviewer for catching this inconsistency.

- **Please specify somewhere that you calculate trial wise parameters.**

As we predict visit-level parameters, we added "visit-level" (lines 59, 61, 71, 106, 172) and a definition of this term in lines 58-59.

- **Why do you not expect "the variance explained to exceed 75 %"? Please elaborate.**

The variability we discuss is stride-to-stride variability across subjects. Since in our work we only have trial-wise gait parameters, our methods require very large amounts of data to be able to predict stride-to-stride variability. Therefore, since stride-to-stride variability explains 25% of total variance in walking speed between patients, we expect our prediction to explain at most 75% of the variance between patients. This, however, could be improved if we had stride-to-stride parameters for each subject. We modified our argument to make this clearer for walking speed (lines 83-85) and GDI (lines 107-110).

- As you discuss variance explained for walking speed, also do that for GDI/GMFCS, please.

In the updated manuscript, we now put the correlation coefficient for GDI in context of reported values (lines 107-110) and tested what features might explain some of the unexplained variability (lines 117-120). We also further discussed variability in GMFCS (lines 112-116).

- Modelling the residuals yields correlations of only 0.18-0.28.

We analyze residuals only to validate if the model can be improved if we added extra information from the frontal plane, in particular foot progression and knee adduction. Model coefficients are significant, confirming this hypothesis. For these analyses R^2 is less relevant and the focus is on the test statistic. We added relevant p-values (line 121). The fact that R^2 is small favors our method, because it suggests that not much can be added to our prediction if we introduce additional signals from the frontal plane.

- Which other parameters did you correlate with the residuals?

We were only interested in the two hypotheses stated in the previous point, showing that our analysis can be improved if one incorporates signals from the frontal plane. There are several other parameters that incorporate frontal-plane information, for example mean pelvic rotation and hip adduction at initial contact. These parameters were correlated with the residuals with $R^2=0.04$ ($p = 0.0001$) and $R^2=0.03$ ($p = 0.001$); see Response Figure 3. However, since these correlations were small, we did not lose much by skipping this information.

Response Figure 3: Two frontal-plane parameters, mean foot progression and mean hip adduction, that partly explain residuals and confirm our hypothesis that some unexplained variability can be addressed with information from the frontal plane.

To further validate our method, we can check if variables visible in the sagittal plane correlate with the residuals of our model. We expect they do not and that is confirmed in our analysis. Here we present results for predicting residuals from mean knee flexion in stance phase of gait and from ankle dorsiflexion at contact. Both are not statistically significant, with $p=0.09$ and $p=0.22$ respectively (Response Figure 4).

Response Figure 4: Two parameters from the sagittal plane, mean knee flexion at stance and ankle dorsiflexion at contact, that do not explain residuals and most likely are implicitly encoded in our prediction.

Finally, we checked if residuals of GDI prediction correlate with age, mass, height, body mass index, cadence, speed, and step length. None of these variables were significant. Given that some of these

variables correlate with GDI, it means that our prediction from video implicitly incorporates these signals.

- Is the result so “significant” that you can argue that it really explains the residual?

For this analysis, our hypothesis was that some of the variance unexplained by our models can be explained with information that OpenPose did not capture from the sagittal plane, such as parameters in the frontal plane. In order to test this hypothesis, we compared residuals of our model with two parameters derived from the frontal plane: foot progression angle and hip abduction angle. The null hypothesis is that these parameters do not correlate with residuals. We got a significant result, allowing us to reject the hypothesis. We conclude that our model could be improved if some of this information was incorporated (for example, by collecting frontal plane videos and computing OpenPose parameters from them). We did not intend to conduct an analysis to show or otherwise imply that these frontal-plane measures fully explain the residuals, nor that this effect is large. We updated the manuscript to make this intention more explicit (lines 117-120).

- You use lab data where the participants wear few clothes. Please discuss how in real world settings at home, clothing would affect the results obtained by OpenPose.

In our datasets, participants are asked to wear minimal, comfortable clothing, including shorts, and walk barefoot. For now, in the real world setting at home, users would be advised to follow these same guidelines. This constraint could be relaxed in the future if data for additional subjects walking in regular clothes were collected. Our results may be already robust to different clothing, but right now there is not enough evidence for this and further validation is needed. The OpenPose algorithm is trained on data from manual annotations, where annotators were asked to approximate positions of centers of joints (defined as intersections of cylinders approximating limbs), even when they were occluded by clothing (Lubomir Bourdev 2011). We are not aware of studies that investigate the accuracy of such human annotations and of OpenPose estimates. We added a comment on clothing and practical deployment of our models in the limitations section (lines 164-166) and Materials & Methods (see Dataset section) of the updated manuscript.

- You give an outlook regarding step width (which would be harder to assess with sagittal plane cameras). In general, please discuss how the viewing direction would affect the results. I imagine that in home settings, you cannot always guarantee a good viewing direction.

We acknowledge two important points in this comment. First, some of the parameters, such as step width or foot progression angle, are very difficult to derive from the sagittal plane. Second, collecting videos in a home setting might be difficult to control.

We agree that for some applications one would need to also collect frontal plane videos to predict parameters visible only in the frontal plane. Our analysis in the response to “Which other parameters did you correlate with the residuals?” reveals, however, that this might not always be necessary. Yet,

collection of the frontal plane could be added to the protocol if necessary. Regarding the second point, for practical use of the current version of the model we would need to ensure that the data collected by the user follows a predefined protocol — our model is based on sagittal plane videos and we would expect this as input.

For the practical commercial deployment and further work, our dataset would have to be extended. In scenarios where one is interested in parameters minimally visible in the sagittal plane, we would suggest designing a protocol in which users collect data from both the sagittal and frontal plane. The training process used in our paper could be replicated, with additional views used as inputs to the model. In order to validate the accuracy for video collected in a home setting, we recommend collecting a new testing dataset that includes videos recorded by naive users, along with gold standard motion capture data. These improvements are beyond the scope of our study, since our focus is on analyzing feasibility rather than building a production-ready deployment.

In the updated version of the manuscript, we discuss this path to deployment together with limitations of our method, lines (lines 164-170).

Supplemental information

- How/where did you show that the change of video systems did not change the predictions of the models? Please elaborate.

In the clinic where the data was collected, there was a change of the room and the camera system in 2004. To check if this change affected our predictions, we analyzed residuals of the model. Our hypothesis was that if the change affected predictions it would be reflected in the mean of residuals or the mean of absolute errors. There was no statistical evidence of that. Here we present results for prediction of GDI, with $p=0.69$ for residuals and $p=0.43$ for absolute errors. Similar results hold for other variables.

Response Figure 5: In order to check if change of the system affected predictions of GDI, we compared residuals (left) and absolute errors (right), before and after the change (vertical black line). There was no significant difference between the means before (red line) and after (blue line) the change.

Given the relatively large sample size of test predictions ($n=348$), this finding reassures us that the bias is negligible.

- **Headline “Hand engineered features” -> “Hand engineered time series” (as features are rather the parameters that you input into the ML models).**

We thank the reviewer for catching this; we’ve changed the headline of this subsection.

- **How many model parameters in total are estimated in your CNN? How does this number compare to the number of input samples/frames for training? Please discuss.**

Our final CNN model has 47,840 trainable parameters and 32,828 samples. We use standard regularization techniques, such as dropout and L2 regularization in order to train the model. These regularization methods are mentioned in the Convolutional Neural Network subsection in Materials & Methods. We added a sentence on the number of parameters in Materials & Methods (Section Convolutional Neural Network, lines 375-377).

While the number of samples is smaller than the number of parameters, our approach is standard practice not only for deep neural networks but also for linear models (for example, in genetics very

often the number of samples is much smaller than the number of genes, but fitting the linear model is still tractable thanks to regularization such as LASSO).

- For RR and RF, you use 12 univariate time series. From the 25 OpenPose keypoints, which one did you use?

We used x and y positions of big toe, ankle, knee, and hip, 2D projections of knee angle and ankle angle, and distance between left and right ankle in the image plane. These are listed in “Extracting Keypoints with OpenPose” in Materials & Methods. However, in the previous version of the manuscript we wrote “e.g., knee, ankle, heel, and first toe” which is misleading since we used explicitly these parameters; it is now corrected in the same subsection.

Reviewer #4 (Remarks to the Author):

The authors present a manuscript on the use of OpenPose to track walking in children with cerebral palsy from laboratory based sagittal plane videos. The tracking of key points is then used as input to 3 different models (convolution neural networks, random forest and ridge regression) to predict gait metrics and likelihood of surgery. The manuscript is clear and follows a logical order. The use of OpenPose and neural networks in the field of gait analysis is novel and there is certainly evidence to suggest that this is an emerging area for the field which will certainly be of interest. The authors have provided enough methodological detail and made their source code available online to allow their experimental setup to be reproduced.

OpenPose was trained on a large varied dataset and is highly likely to be applicable to this patient population. While the authors claim these methods would allow for at home clinical gait analysis on a smart-phone I find this claim overstated. The videos used in this study are from a laboratory setting, with experienced personnel, the camera on a tripod tracking the subject from the side, there is a clean background and the patient in wearing minimal clothing, exposing the body parts of interest. Videos taken in the 'real world' would differ greatly. It would be more reasonable to state that this approach would provide a tool for monitoring progression or managing time of referral to appropriate services.

We agree that the videos taken in the laboratory differ from videos taken at home. Even if we assume that we require users to exactly replicate the protocol at home (i.e. have the same distance to walk in a straight line and operate a camera from exactly the same distance to that line of walking) other artifacts might make the analysis inaccurate.

We softened the claims regarding the use of our models explicitly at home by changing the title, explaining in detail which parameters we can track (lines 58-62), and adding a paragraph on practical use and limitations (lines 163-180). However, we believe that our methodology and article include enough evidence that both software and hardware is at a stage where deriving clinically-relevant motion metrics using a commodity camera is possible. At-home use would require a substantial investment in engineering and further research, which we make clear in the revised paper. In particular, we highlighted that in practice users would either need to follow a strict protocol for camera-based measurements or more data would have to be collected to cover variability of conditions (lines 164-168).

Further consideration should be given to the reporting of the statistical analysis. In the text, only the correlations of model predicted metrics with ground truth has been reported, no measure of variance has been included, although displayed in the figures. This is relevant especially as the authors are looking to explain the variance in the results.

We thank the reviewer for pointing this out since proper analysis of variance is essential to our analysis. Given the stochasticity of the estimates of our statistics, particularly the correlation coefficient, it is desirable to report their variability.

In the updated version of the manuscript, we addressed the first point by adding information about the confidence intervals of our estimators. These were previously shown visually in Figure 2, but we added the details in Table 1.

Most of the analysis of variance can be directly derived from the coefficient of determination r reported in the text. We updated our analysis of variance for walking speed (lines 83-85) and GDI (lines 106-110 and 117-122). We added more details in Materials & Methods on how variance can be decomposed for our measurements, assuming a nested model, i.e., multiple trials in multiple visits of multiple patients (Section Evaluation in Materials & Methods).

Prediction of longitudinal change in gait metrics requires further consideration. The correlations found for those that underwent SEMLS is clearly influenced by the point at the beginning of the graph. There are also likely interactions between terms that need to be taken into consideration.

We appreciate this point as it led us to a more thorough analysis of longitudinal changes. First, the context where we anticipate these predictions might be of importance is when subjects visit the clinic to get precise measurements and then use a video camera to track changes after the visit. To that end, when we model change in GDI we can incorporate raw predictions and the initial clinical GDI. Second, instead of building separate models for patients who did or did not undergo SEMLS, we can include SEMLS as a covariate. Such a framework allows us to test for interaction effects between SEMLS and the predicted change. Third, we can also check if other variables, such as age or GMFCS, improve the prediction, since they are also measured in the clinic at baseline.

This updated analysis allows us to predict the change with higher accuracy than previously reported. While the task has changed, we believe that the current version better illustrates practical applicability, particularly as a tool that is complementary to clinical practice. In the updated analysis, SEMLS and the interaction effect between SEMLS and the predicted difference are not significant in the model — the only two significant predictors are the predicted difference and the baseline GDI. In the updated manuscript we described the new approach and results (lines 134-141).

Mismatch of results, graph for walking speed correlation 0.73, text states 0.75.

We thank the reviewer for pointing that out; we corrected the number (lines 12, 76, 81).

References

- Bland, J. Martin, J. Martin Bland, and Douglas Altman. 1986. "STATISTICAL METHODS FOR ASSESSING AGREEMENT BETWEEN TWO METHODS OF CLINICAL MEASUREMENT." *The Lancet*. [https://doi.org/10.1016/s0140-6736\(86\)90837-8](https://doi.org/10.1016/s0140-6736(86)90837-8).
- Fasel, Benedikt, Cyntia Duc, Farzin Dadashi, Flavien Bardyn, Martin Savary, Pierre-André Farine, and Kamiar Aminian. 2017. "A Wrist Sensor and Algorithm to Determine Instantaneous Walking Cadence and Speed in Daily Life Walking." *Medical & Biological Engineering & Computing* 55 (10): 1773–85.
- Lin, Tsung-Yi, Michael Maire, Serge Belongie, James Hays, Pietro Perona, Deva Ramanan, Piotr Dollár, and C. Lawrence Zitnick. 2014. "Microsoft COCO: Common Objects in Context." *Computer Vision – ECCV 2014*. https://doi.org/10.1007/978-3-319-10602-1_48.
- Lubomir Bourdev, Jitendra Malik. 2011. "The Human Annotation Tool." June 17, 2011. <https://www2.eecs.berkeley.edu/Research/Projects/CS/vision/shape/hat/>.
- Luo, Zelun, Jun-Ting Hsieh, Niranjan Balachandar, Serena Yeung, Guido Pusiolo, Jay Luxenberg, Grace Li, et al. 2018. "Computer Vision-Based Descriptive Analytics of Seniors' Daily Activities for Long-Term Health Monitoring." *Machine Learning for Healthcare (MLHC) 2*.
- Rasmussen, Helle Mätzke, Dennis Brandborg Nielsen, Niels Wisbech Pedersen, Søren Overgaard, and Anders Holsgaard-Larsen. 2015. "Gait Deviation Index, Gait Profile Score and Gait Variable Score in Children with Spastic Cerebral Palsy: Intra-Rater Reliability and Agreement across Two Repeated Sessions." *Gait & Posture* 42 (2): 133–37.
- Sato, Kenichiro, Yu Nagashima, Tatsuo Mano, Atsushi Iwata, and Tatsushi Toda. 2019. "Quantifying Normal and Parkinsonian Gait Features from Home Movies: Practical Application of a Deep Learning-based 2D Pose Estimator." *PLOS ONE*. <https://doi.org/10.1371/journal.pone.0223549>.
- Seethapathi, Nidhi, Shaofei Wang, Rachit Saluja, Gunnar Blohm, and Konrad P. Kording. 2019. "Movement Science Needs Different Pose Tracking Algorithms." <http://arxiv.org/abs/1907.10226>.
- Xue, David, Anin Sayana, Evan Darke, Kelly Shen, Jun-Ting Hsieh, Zelun Luo, Li-Jia Li, N. Lance Downing, Arnold Milstein, and Li Fei-Fei. 2018. "Vision-Based Gait Analysis for Senior Care." <http://arxiv.org/abs/1812.00169>.
- Xu, Xu Xu, Raymond W. McGorry, Li-Shan Chou, Jia-Hua Lin, and Chien-Chi Chang. 2015. "Accuracy of the Microsoft Kinect™ for Measuring Gait Parameters during Treadmill Walking." *Gait & Posture*. <https://doi.org/10.1016/j.gaitpost.2015.05.002>.

Reviewers' Comments:

Reviewer #3:

Remarks to the Author:

Thank you for submitting the very much improved version of your manuscript with explanations related to the previous major concerns, regarding:

1) Visit-level parameters:

Considering your argumentation that visit-level gait parameters are sufficient for obtaining clinically meaningful mobility estimates for the respective population, the validation seems plausible (contrary to my previous assessment). As major improvement compared to the previous version of the manuscript, this is now well discussed in the Introduction and Discussion.

2) The additional analyses regarding the comparison of OpenPose + DL versus other methods highlight the relevance of using DL models as compared to other methods. This point has been well addressed.

3) The additional final paragraph makes it easier for the reader to comprehend the main limitations of the study.

In general, the main points of criticism of the manuscript have been well addressed throughout the paper. All points have been addressed in either manuscript or clarified in the direct response to the reviewer.

Some minor additions (arising from the checklist documents) are needed before publication:

Manuscript:

- Please add a note that informed consent was obtained by the participants (or refer to the original study)

Code repository (github): Please...

- describe in the Readme.md the order of how the notebooks need to be run (or add s.th. like a main / demo file)
- add dependencies including version numbers (they appear in one of the checklists, but should also be mentioned in the repository)
- add a small dataset for a demo + a demo
- add instructions on how to run on new ('your') data

Reviewer #4:

Remarks to the Author:

The authors present a revised manuscript on the use of OpenPose to track walking in children with cerebral palsy from laboratory based sagittal plane videos. The tracking of key points is then used as input to 3 different models (convolution neural networks, random forest and ridge regression) to predict gait metrics and likelihood of surgery.

The authors have addressed my questions and made considerable changes to the manuscript. I have a few minor questions and recommendation for the manuscript that can be found in the attached annotated PDF.

Response to referees letter: Deep neural networks enable quantitative movement analysis using single-camera videos (NCOMMS-20-03494A)

We thank reviewers for a careful and diligent review of our updated manuscript. We updated the text following their advice.

Reviewer #3

Please add a note that informed consent was obtained by the participants (or refer to the original study)

The note about informed consent is included in the Dataset subsection in Methods.

Code repository (github): Please...

- describe in the Readme.md the order of how the notebooks need to be run (or add s.th. like a main / demo file)

We list notebooks in the order in which they should be run and we now mentioned that in the README file

- add dependencies including version numbers (they appear in one of the checklists, but should also be mentioned in the repository)

We exported all dependencies with the exact version to the requirements.txt file. In the readme file we mentioned that our software requires python 3.7 and demo also requires NVIDIA docker (to run OpenPose).

- add a small dataset for a demo + a demo

We now prepared a sample video and it's processed version (so that a user doesn't have to run OpenPose) in the demo directory.

- add instructions on how to run on new ('your') data

Instructions to run a demo are now listed in this file and they are referred to from the main README.

Reviewer #4 Minor comments highlighted in the manuscript by the reviewer:

I agree that these metrics [average walking speed, cadence, and knee flexion angle at maximum extension] are commonly used for diagnostics and planning. However, this leads the reader to believe that decisions can be made only with this information.

Additional information is used for decisions (line 79).

We agree with that statement and made this point more clearly in the manuscript.

What about errors in your model? (line 85).

Our intention was to discuss only Intrantra- and inter- patient variability (not errors) and describe errors in the later part of the paragraph. We have revised the paragraph to reflect this.

The authors attribute error in predicting gait metrics to stride-to-stride variability given video and 3D were performed separately but within the same session. While this may account for some error it is likely this is due to the models inability to predict this metric with high accuracy (line 93).

We also now explicitly state that the unexplained error is due to the model's error.

This [Low correlation of OpenPose estimate of the knee flexion angle at maximum extension] is also because the camera is not positioned directly square with the patient throughout the recording. Therefore, keypoints directly from OpenPose are affected (line 99).

While we are unable to collect data explicitly confirming this hypothesis, we added an appropriate note in that paragraph highlighting that the error might be related to our data collection technique and not only OpenPose itself.

This correlation [between GMFCS assessed by experts and parents] is low as parents are not trained GMFCS (line 118).

We agree with the reviewer that parents are not trained to assess GMFCS and their correlation with experts is low. However, we bring this number up in order to show that parents might be unable to identify impairments in their kids. In the Discussion section, we suggest that our tool can potentially help parents to early identify walking problems (line 163-164).

This [enabling fast and easy quantitative motion analysis in a natural environment] is still over stated. Shows potential for use of video based pose estimation to predict gait metrics. (line 189).

We have now softened the language to address this concern.

How many [videos] were excluded because OpenPose failed to detect patients accurately. State initial number of videos (line 248)

As the reviewer requested we now state the number of videos disregarded in our filtering process.